# Hydride formation pressures and kinetics in individual Pd nanoparticles with systematically varied levels of plastic deformation

Carl Andersson [1], Jonathan Zimmerman [2], Joachim Fritzsche [1], Eugen Rabkin [2] ✉ & Christoph Langhammer [1] ✉

Pd nanoparticles, together with bulk and thin film Pd, constitute the archetype model system for metal-hydrogen interactions. The density of defects in Pd nanoparticles, such as grain boundaries and dislocations, combined with their size, shape, composition and lattice strain, dictate their hydrogen sorption kinetics and thermodynamics. Despite decades of research and its relevance in applications, such as solid-state hydrogen storage, hydrogen sensors, hydrogen embrittlement, and hydrogen separation membranes, a coherent picture of the intricate interplay between defects, strain and Pd nanoparticle hydrogen sorption properties is missing. Here, we employ a combination of single particle nanocompression, single particle plasmonic nanoimaging and high-resolution cross-sectional single particle TEM imaging to investigate hydrogen absorption kinetics and hydride phase formation pressures in a nanofabricated array of Pd nanoparticles on sapphire substrate with systematically varied levels of plastic deformation – and thus defects and strain. We not only show a clear deformation-level dependent trend of both the kinetics and the hydride formation pressure, but also reveal their complex evolution upon hydrogen cycling. We discuss how these results provide a quantitative view of the impact of plastic deformation on nanoscale metal hydrides, and how they reveal the surface and bulk morphology of Pd nanoparticles upon repeated hydrogen cycling.

Structural transformations of a crystalline lattice accommodating solute atoms, such as hydrogen during hydride formation in metals, is accompanied by lattice strain, dislocation generation/annihilation, and plastic deformation. Accordingly, a significant body of literature discusses these aspects for bulk and thin film systems, with Pd in focus due to its distinct role as the model system for metal hydrides[1–4]. For Pd nanoparticles on the other hand, the smallest Pd-hydrogen system one can effectively study, the literature is sparser. The key conclusion of the few previous studies is that for dislocation-free single crystalline Pd nanoparticles, the hydride formation thermodynamics[5–7] and kinetics[8] become particle size dependent, since the lattice expansion-related energy barrier for metal-hydride transformation scales with the system volume[9]. This ideal behavior breaks down as soon as dislocations either are present in the crystal from the beginning or are nucleating during hydrogenation, thereby enabling plastic deformation by the motion of dislocations, to alleviate the volume-expansion-induced lattice strain[10–13].

[1]Department of Physics, Chalmers University of Technology; 412 96, Gothenburg, Sweden. [2]Department of Materials Science and Engineering, Technion - Israel Institute of Technology, 320000 Haifa, Israel. ✉e-mail: erabkin@technion.ac.il; clangham@chalmers.se

As a second aspect, metal hydride nanoparticles used in practical applications are universally complex defect-rich structures. This is the consequence of their preparation methods, such as high-energy ball-milling, valued for its ability to impose altered hydride phase transition pressures and hydrogen absorption kinetics, two crucial parameters in applications of hydride systems[14–17]. The fundamental problem is that the complex structures created make deconvolution of the mechanistic origins of their advantageous properties effectively impossible. For example, there is currently no method to distinguish between the effects of particle volume (and thus diffusion path) reduction and the creation of dislocation networks, since both are intertwined in nanoparticles created using high-energy ball-milling. Furthermore, the hydrogenation process itself may alter the dislocation networks, e.g., by creating new dislocations[13,18–22] or annihilating pre-existing ones[12,23]. It may also introduce or alter other types of defects, such as stacking faults[24] and 9R-phases[25]. This results in highly complex internal microstructure, where the hydrogenation properties of a hydride forming metal like Pd are influenced not only by the initial strain and defect state of the system, but also by its hydrogenation history.

As a consequence, there is a lack of systematic investigations of how plastic deformation—and thereby dislocations—impact the hydrogen sorption process in metal hydride nanoparticles in general, and in Pd nanoparticles in particular. To fill this gap, we present here a study of an array of crystalline Pd nanoparticles nanofabricated onto sapphire substrate, whose level of initial plastic deformation we control at the single particle level by means of systematically varied degrees of nanocompression[26–29]. Subsequently, we assess the hydrogenation kinetics and thermodynamics of a total of 56 individual nanoparticles with different levels of nanometer precise nanocompression together with 289 non-compressed control particles simultaneously, in a single experiment, using multiplexed single particle plasmonic nanoimaging[12,30,31], followed by electron microscopy analysis of their morphology. We find a systematic dependence between the degree of deformation, the hydrogen absorption kinetics and hydride phase formation pressure, with the more compressed particles displaying faster kinetics and lower hydride formation pressures. Furthermore, we observe that compressed particles retain their fast hydrogenation kinetics much longer than non-compressed ones during hydrogen cycling. All particles also show a cyclic evolution of their hydride formation pressures. As we demonstrate, this behavior is mediated by differing evolutions of the dislocation structure at the surface versus the bulk of the Pd nanoparticles. We also emphasize here that, following the discussion above regarding the difficulty of full reconstruction of the individual defect networks of hundreds of nanoparticles, in this work, we will keep the discussion of different types of defects non-specific and mainly focus on general deformation trends. We also note that both stacking faults and patches of the 9R phase are formed as a result of nucleation and propagation of Shockley partial dislocations, such that a general discussion in terms of dislocation substructure is still relevant.

## Results and discussion
### Sample preparation and nanoindentation
For our study, we nanofabricated a regular array of a total of 345 Pd nanoparticles onto a sapphire substrate by means of electron-beam lithography (EBL) - (Fig. 1a). The disk-shaped particles had a nominal diameter of 200 nm and a thickness of 60 nm directly after fabrication and were placed 10 μm apart in the array to enable their individual detection in single particle plasmonic nanoimaging. After annealing in 2% $H_2$ in Ar atmosphere at 500 °C for 2 h, this yielded crystalline particles with an abundance of dislocations throughout the bulk, as well as a high concentration of dislocations in the interface region towards the substrate (Fig. 1b). The particles are either polycrystals comprised of a few grains separated by high-angle grain boundaries or single crystals, both with a mean diameter of ~ 160 nm (Fig. 1c–e and SI Fig. 1).

Subsequently, we compressed a sub-fraction of these particles in this array using a nanoindenter equipped with a flat diamond punch, while simultaneously recording the corresponding single particle load-displacement curves (Fig. 1a, f). Specifically, we compressed sub-groups of 7 particles each to nominally 5 nm, 10 nm, 15 nm, 20 nm, 25 nm, 30 nm, 35 nm and 40 nm below their nominal thickness (Fig. 1g–i). The correspondingly obtained load-displacement curves (see SI Fig. 2) can be grouped into two classes: i) the strain-burst type[28,32] (Fig. 1j), characteristic for defect-free single crystalline Pd particles. It can be understood as a homogeneous dislocation nucleation when the local shear stress reaches the yield strength of the crystal[26–29]. ii) The staircase-yielding type[33] (Fig. 1k), characteristic for a particle containing grain boundaries, dislocations and/or their sources. In such particles, the onset of plasticity is associated with easy dislocation nucleation at grain boundaries, activation of dislocation sources, or movement of pre-existing dislocations, thus alleviating the need for one single strain burst. To this end, it was demonstrated in experimental studies and atomistic simulations that a singular strain burst during microcompressions of pristine single crystalline metal nanoparticles is associated with an avalanche-type massive nucleation of dislocations at the particle-punch interface, typically at surface facet edges and corners[26,27,29,34,35]. This concerted dislocation nucleation can also manifest itself as an abrupt load drop in the case the compression test is performed in displacement-controlled mode[36]. A transition from the strain burst to staircase yielding behavior has been observed after controlled introduction of defects into pristine nanoparticles, e.g., by Ga+ ions irradiation in a focused ion beam (FIB) instrument[37]. Interestingly, metal nanostructures with a high density of dislocations can experience a reverse transition from the staircase to strain burst behavior in a process known as mechanical annealing: the pre-existing dislocations glide towards the free surfaces and annihilate there upon compression, leaving behind a pristine structure[38,39]. From the yield strength of the particles, we can estimate that the dislocation density in the non-deformed annealed particles is about $10^{11}$-$10^{12}$ m$^{-2}$, which would put the annealed (but not deformed) particles in the dislocation starvation regime[40], where the pre-existing dislocations glide to and annihilate at the surface at increased stress levels, e.g., during hydrogenation (see SI section 3 for more information). Also, we estimate the post-deformation dislocation density in the compressed particles to be of the order of $10^{15}$–$10^{16}$ m$^{-2}$ (see SI section 4). In addition to revealing the pre-indentation microstructure (i.e., whether a particle is single- or polycrystalline), the load-displacement diagrams (SI Figs. 2–3) also provide information about the load corresponding to a given punch displacement.

### Hydrogen absorption kinetics measurements
In the next step, we mounted the sample in a (vacuum) chamber with an optical window to enable both rapid change of the sample environment from vacuum (base pressure 1 μbar) to a specific $H_2$ pressure (the filling time constant of the chamber is approx. 1 s for a 350 mbar $H_2$ pulse) in $H_2$ absorption kinetics experiments, and to enable slow step-wise in/ decrease of the $H_2$ pressure in Ar carrier gas in optical pressure-composition isotherm measurements (SI Fig. 6). This chamber was mounted onto an upright optical microscope equipped with a reflection-mode dark field objective and an imaging EMCCD camera to enable plasmonic nanoimaging measurements with single particle resolution during hydrogenation of the sample (see Methods).

We then exposed the sample to a first 350 mbar $H_2$ pulse and recorded the change in light scattering intensity over time for all particles. We note that the scattering intensity change is proportional to the amount of hydrogen absorbed by a particle[41–43]. Selecting first one particle representative of each compression sub-group and plotting the normalized scattering intensity as a function of time at the exposure to the very first $H_2$ pulse, we notice a distinct compression-level dependence of the response, where the most compressed

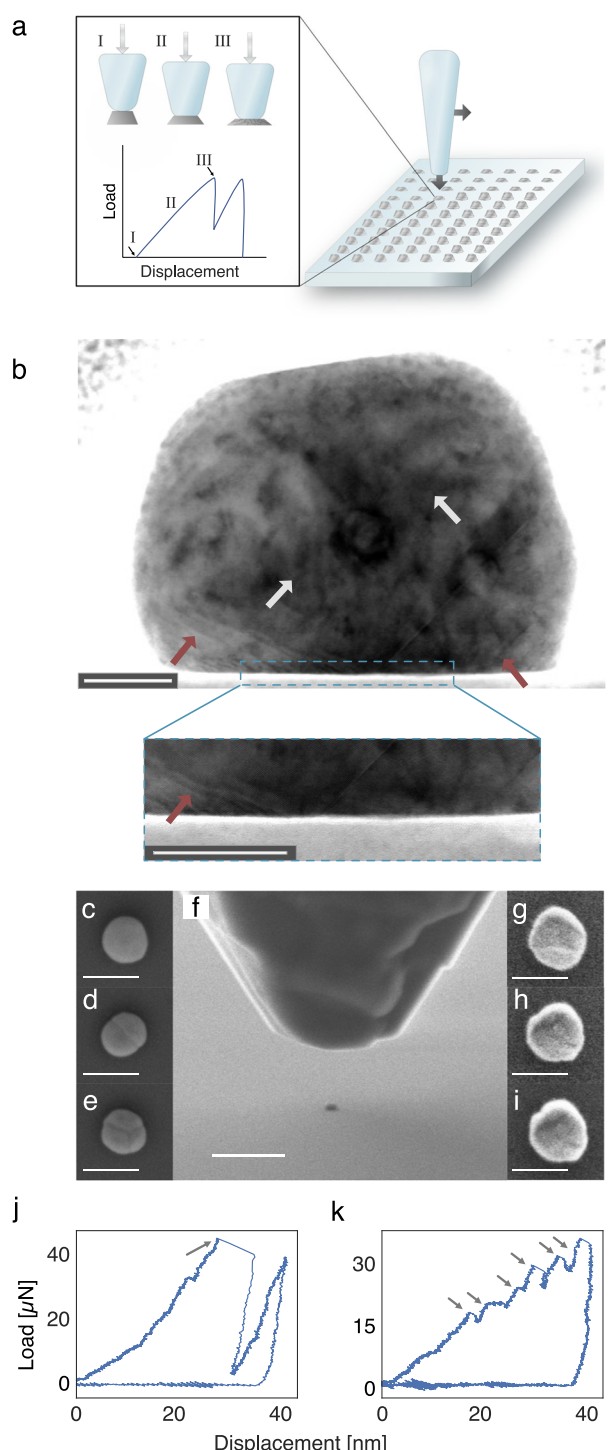

**Fig. 1 | Nanocompression of single Pd nanoparticles. a** Schematic depiction of the nanocompression process of individual particles in a Pd nanoparticle array using a nanoindenter equipped with a flat diamond punch. A representative load-displacement curve for a single crystalline defect-lean particle is shown and three key stages are highlighted: I) contact with the particle, II) elastic deformation and III) plastic deformation characterized by a sudden drop of the load, which corresponds to a strain burst when the local shear stress reaches the yield strength of the crystal. **b** Bright-field transmission electron microscope (TEM) image of a Pd nanoparticle cross section obtained from an as-fabricated particle prior to the first $H_2$ exposure. The particle cross-section lamella was produced using a focused ion beam (FIB—see "Methods"). It reveals the crystalline and well-faceted nature of the particle, as well as the presence of dislocations corresponding to the dark areas in the image, two of which are highlighted by the white arrows. The image, and in particular the zoom-in, also reveal a high density of dislocations in the particle-substrate interface region (red arrows), which hint at significant clamping strain. Scale bars in both images are 20 nm. A single lamella cross section was fabricated from one of 345 representative particles. **c**–**e** High-resolution SEM micrographs of representative Pd nanoparticles before compression and any exposure to hydrogen. We note the different number of grains and varying grain sizes, even though the particles are from the same sample. The particle in (**c**) is a defect-lean single crystal whereas the particles in (**d**) and (**e**) are either polycrystals or defect-rich single crystals. Scale bar is 200 nm. **f** SEM micrograph of the diamond punch of the nanoindenter above a Pd nanoparticle, just before compression. Scale bar is 1 μm. **g**–**i** High-resolution SEM micrographs of representative Pd nanoparticles after compression (not the same particles as in **c**–**e**) but before any exposure to hydrogen. We note the significant morphological change of the particles after nanocompression. Scale bar is 200 nm. **j, k** Representative load-displacement curves for two individual nanoparticles. The left one (**j**) exhibits the characteristics of a strain burst of a single crystal (arrow), whereas the right one (**k**) exhibits a stepwise staircase-yielding (arrows), which is characteristic for a polycrystalline particle.

This is generally in line with polycrystalline thin film experiments that exhibited faster hydrogen absorption rates upon increased plastic deformation and correspondingly increased dislocation density[44].

To further analyze the response of the nanoparticles, we organized them into groups represented by their actual compression level (the actual compressions deviate slightly from the nominal compression areas in Fig. 2b, see SI Fig. 3), as revealed by the individual load-displacement diagrams for each particle (SI Fig. 2). From here on, only the actual compression levels will be used. Analyzing the average response for each compression sub-group corroborates the clear trend of decreasing $t_{30\text{-}50}$ for increasing compression (Fig. 2c). Furthermore, more detailed inspection reveals a significant spread in the single particle $t_{30\text{-}50}$ values of up to 5 s within each sub-group (first standard deviation). Tentatively, we ascribe this spread, which is equally significant for compressed as for as-fabricated (non-compressed) particles in this first hydrogenation cycle, to inherent morphological differences, such as the intrinsic single- or polycrystallinity[11] for the as-fabricated particles, and slightly different levels of compression, and thus plastic deformation, for the compressed particles.

Upon further cycling by exposing the sample to repeated 350 mbar $H_2$ pulses, we observed that after two cycles the $t_{30\text{-}50}$ values are reduced across the board, and that simultaneously the trend between the different compression-level sub-groups is retained (Fig. 2d). This overall faster response is even more pronounced after cycle 6, while the inter-sub-group trend still is retained (Fig. 2e). Interestingly, not only the absolute $t_{30\text{-}50}$ values are reduced between cycle 1 (Fig. 2c) and cycle 6 (Fig. 2e), but also the spread between individuals within each compression sub-group. The accelerated kinetics together with the reduced spread in $t_{30\text{-}50}$ is most likely the result of the reduction of surface oxide layers formed upon exposure to ambient conditions after fabrication (for further discussion regarding why oxides or other surface contaminants should not have any significant effect on the results, see SI section 6). After 18 cycles, the accelerated response, as well as the narrow $t_{30\text{-}50}$ distribution, is

particle hydrogenates the fastest and the as-fabricated (non-compressed) one the slowest, with the intermediate particles systematically placed in between these two extremes (Fig. 2a). This trend is corroborated in the response time, $t_{30\text{-}50}$, defined as the time between 30% and 50% signal intensity change after $H_2$ exposure, in the color-coded dark-field scattering image of the entire array (Fig. 2b). It reveals that indeed the nominally most strongly compressed particles absorb $H_2$ the quickest and that $t_{30\text{-}50}$ increases systematically for lower levels of compression, being the highest for the non-compressed particles.

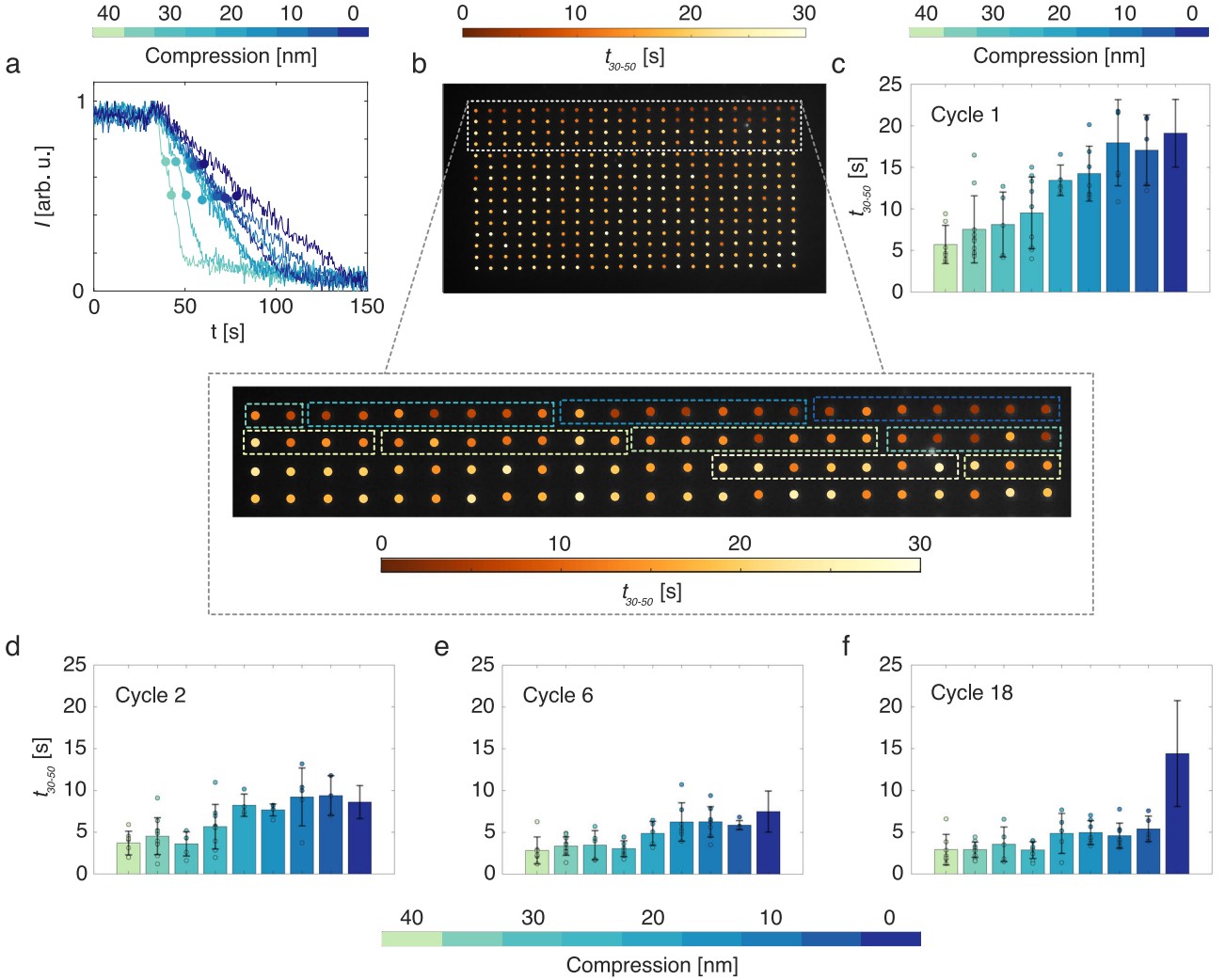

**Fig. 2 | Hydrogen sorption kinetics of single compressed and non-compressed Pd nanoparticles. a** Normalized dark-field scattering intensity time traces for representative single particles for each of the nine different compression degrees. The absorption times are quantified by $t_{30\text{-}50}$, which is defined as the time between 30% and 50% signal intensity change after hydrogen exposure. The 30% and 50% signal intensity change points are each marked with a filled circle on every individual time trace. **b** Dark-field scattering image of the nanoparticle array. The particles are color-coded with their individual $t_{30\text{-}50}$ values for their first exposure to hydrogen after nanocompression (cycle 1). The area with the compressed particles is highlighted with a dashed rectangle and is shown in magnified form where the nominal compression-levels of the particles are outlined with a dashed box with a color representing their degree of compression. **c** Histogram of the average absorption time, $t_{30\text{-}50}$, during cycle 1 for the 9 different compression-level sub-groups (sorted by their actual compression level derived from the individual load-displacement curve of each particle, which deviate slightly compared to the nominal compressed areas in (**b**), see SI Figs. 2–3). The error bars indicate one standard deviation calculated from the individual $t_{30\text{-}50}$ absorption times of the particles within each compression-level sub-group. For the compressed particles, the individual data points are also plotted as filled circles. **d**–**f** Histograms of the average absorption time $t_{30\text{-}50}$ for cycle 2 (**d**), cycle 6 (**e**) and cycle 18 (**f**).

retained for all compressed particles, while the non-compressed particles have started to significantly slow down and develop a wide $t_{30\text{-}50}$ distribution (Fig. 2f).

To further investigate the evolution of the $H_2$ sorption response of the differently compressed particles, we continued the hydrogen cycling up to 70 cycles. As the first finding, we note that beyond cycle 18, the non-compressed particles keep decelerating to a quite extreme level. This distinctly sets them apart from the compressed ones, which also continuously decelerate but at a much lower rate (Fig. 3a). Interestingly, we also see that the distinct compression-level dependence of the response described earlier, where the most compressed particles hydrogenate the fastest and the non-compressed ones the slowest, is retained all the way to cycle 70.

During the entire hydrogen cycling sequence, we inserted a total of sixteen pressure-composition isotherm measurements (SI Fig. 8, nine for absorption, seven for desorption) after absorption kinetics cycles 3, 15, 27, 39, 51 and 63, and as cycles 73–76, to also assess the

impact of nanocompression on hydride formation thermodynamics. In this type of measurement, the particles were exposed to stepwise slowly increasing $H_2$ concentration in Ar carrier gas (300 s per concentration step, for details see Methods) at constant temperature and atmospheric pressure. In this way, the sample is given enough time to absorb the amount of hydrogen corresponding to thermodynamic equilibrium for each $H_2$ partial pressure. We thus resolve the first-order phase transformation from the hydrogen-poor solid solution (α-phase) to the hydrogen-rich hydride (β-phase), which is characterized by a distinct two-phase coexistence "plateau", as well as by distinct hysteresis upon its reverse, in the resulting optical pressure–composition isotherms (SI Fig. 8). To reduce drift, these long measurements were split in two parts, where absorption isotherms were measured first. Importantly, during each isotherm measurement, the particles spent ~ 6 h continuously in $H_2$ environment and thus in a hydrogenated state. This is ~72 times longer than during a kinetics cycle, which is only 5 min long. This aspect is of special interest since we clearly

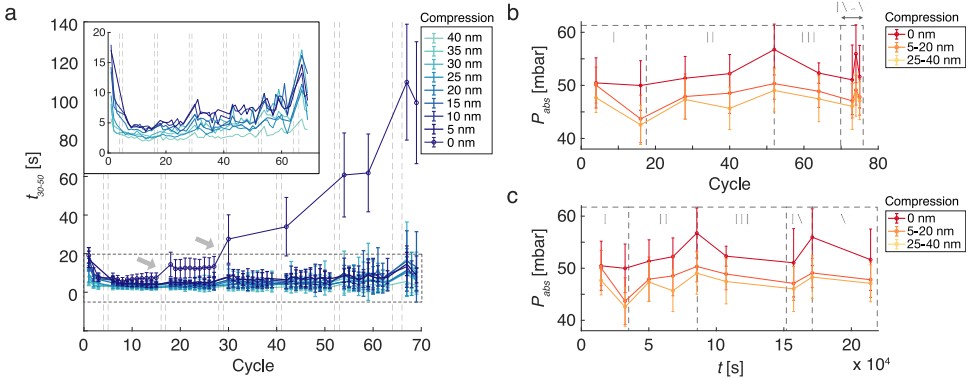

**Fig. 3 | Hydrogen absorption time and absorption plateau pressure as function of hydrogenation cycles. a** Evolution of the average $t_{30-50}$ absorption times for the 9 compression-level sub-groups as a function of hydrogenation cycles. Cycles where isotherms were measured instead of kinetics are highlighted with vertical dashed lines. The error bars indicate one standard deviation calculated from the individual $t_{30-50}$ absorption times of the particles within each compression-level sub-group (see SI Figs. 2–3 for the number of particles in each sub-group). We note the extreme deceleration of the absorption kinetics for the non-compressed particles (0 nm compression) in later cycles. We also note that the greatest deceleration happens directly after isotherm measurements. This is most apparent after the second and third isotherm measurement (highlighted with gray arrows). Due to the significant deceleration of the non-compressed particles, only cycles where at least 200 of these particles have completely desorbed before every new kinetics measurement are shown. For more information, see SI section 10. Inset: Magnification of the average absorption times for the 5–40 nm compressed particles (area outlined with a dashed, gray box in the main figure) showing a consistent,

compression-level-dependent spread in $t_{30-50}$ values. **b, c** Evolution of the average absorption plateau pressures $P_{abs}$ as a function of hydrogenation cycle (**b**) and as a function of time spent in > 40 mbar $H_2$ (**c**). The particles are divided into three groups depending on their degree of compression, i.e., non-compressed (0 nm), 5–20 nm compression and 25–40 nm compression. The error bars indicate one standard deviation calculated from the individual absorption plateau pressures for the particles within each compression-level sub-group (see SI Figs. 2–3 for the number of particles in each sub-group). To have comparable results between the compressed (5-20 nm and 25–40 nm group) and the non-compressed (0 nm) groups of particles, only particles in the non-compressed group (0 nm) with hydrogen absorption times comparable to the particles in the compressed groups have been used in (**b, c**). For more information, see SI section 11. The evolution of the average absorption plateau pressures $P_{abs}$ is divided into five phases (I-V), which are described in more details in SI section 12. For the $P_{abs}$ evolution of all individual compression-level sub-groups, see SI Fig. 13.

observe that both compressed and non-compressed particles exhibit significant deceleration after each isotherm measurement, with the effect being most pronounced for the non-compressed particles (Fig. 3a, and see also SI Fig. 9).

We argue that this effect relates to hydrogen-induced increased dislocation mobility in the hydride[18,21,22,45]. Accordingly, the long and continuous dwell in a hydrided state during an isotherm measurement explains why the sorption deceleration is more pronounced after isotherms compared to a preceding series of (much shorter) $H_2$ pulses. This hypothesis, however, also requires that the non-compressed particles, both poly- and single crystalline, have pre-existing dislocations, despite them not being compressed and thus plastically deformed. This is clearly observed in the TEM images of a cross section of a non-compressed particle prior to the first exposure to $H_2$ (cf. Figure 1b). This argument is further corroborated by extrapolating the scaling law for hydrogen absorption kinetics in defect-free single crystalline particles[8] to the sizes of the particles in this study, which reveals that our particles in the first cycle indeed hydrogenate much faster than what would be expected from a perfect defect/dislocation-free single crystal (SI Fig. 10).

Having confirmed the initial presence of dislocations and their role in accelerating hydrogen sorption[12,44,46–48], we can explain the observed trends in absorption kinetics as follows. In the non-compressed particles, which feature fewer (in the polycrystals) or much fewer (in the single crystals) dislocations than the compressed ones, existing dislocations can reach the surface and shear the crystal upon hydrogenation (SI Fig. 1b). This process causes the dislocations to disappear, leading to a gradual decrease in dislocation density. As a result, hydrogen absorption slows down over time in non-compressed particles, since a high dislocation density accelerates hydrogen absorption[44]. A close analogy is thermal annealing, which is well known to reduce the dislocation density in Pd thin films[46] and to slow down hydrogenation kinetics in annealed particles[12]. Also, some similarity can be drawn to mechanical

annealing, whereas plastic deformation expels pre-existing dislocations from a nanocrystal[39].

For the compressed particles on the other hand, we propose that nanocompression induces a larger initial dislocation density compared to the as-fabricated particles. Indeed, dislocation densities as high as $2 \times 10^{16}$ $1/m^2$ were reported in compressed Cu nanoparticles[49], and formation of sessile dislocation structures after nanoindentation of Pt nanoparticles has also been observed[50]. We hypothesize that such sessile dislocation structures can survive the process of hydrogen cycling (see SI section 13 for more information). This explains why the compressed particles hydrogenate faster– either because the hydrogenation strain barrier[9,51] is reduced, and/or because high dislocation density enhances hydrogen diffusion[44,46]–and why they retain this kinetic advantage over many hydrogenation cycles.

To put the kinetics enhancing effect of the plastic deformation into perspective, we note that the difference between the average absorption time for the compressed vs. the non-compressed particles in the final cycles is essentially one order of magnitude. In comparison, alloying Pd with Au, a widely established method for accelerating hydrogen absorption kinetics in nanostructured Pd, also accelerates kinetics approximately by an order of magnitude[52–55], thereby showcasing the importance of defect structures and morphology in tuning the overall kinetics of the particles.

## Hydrogen absorption isotherms

To corroborate that plastic deformation governs the distinct deformation-dependent kinetic behavior of the different particle sub-groups, it is useful to analyze the corresponding isotherm measurements, which provide indirect insight into their strain state. Specifically, we extract the plateau pressures for the absorption, $P_{abs}$, branch (SI Fig. 8), and plot how this parameter evolves with $H_2$ cycling (Fig. 3b). We also note that according to the Schwarz-Khachaturyan-Griessen (SKG) model, $P_{abs}$ for an ideal Pd system is directly related to the strain state of the system[9,10,56]. This state has three main

contributions: (i) the internal tensile strain caused by the volume expansion due to hydrogen occupying interstitial lattice sites; (ii) external compressive strains induced by a strong interaction with the substrate—so called clamping effects—which are significant in thin films or nanoparticles attached to solid substrates. These effects likely explain the high dislocation density observed in the interface region of non-compressed particles prior to $H_2$ exposure (cf. Figure 1b); (iii) other internal strains stemming from defect networks, such as dislocations and low-angle grain boundaries[46,53,57,58]. Indeed, for all cycles, we see a higher $P_{abs}$ for the non-compressed particles compared to their compressed counterparts (Fig. 3b), implying that the plastic deformation induced by the nanocompression acts to relieve compressive stress in the deformed particles, and as such leaves these particles with a reduced energy barrier for the α-to-β phase transition. Additionally, comparing the least compressed group (5–20 nm compression) to the most compressed group (25–40 nm compression) apparently reveals that the greater the compression level, the greater reduction in $P_{abs}$. However, this difference is not statistically significant (see SI section 14 for a statistical analysis).

To put the observed reduction of $P_{abs}$ into perspective, i.e., an average decrease of $\sim 4$ mbar for the 5–20 nm compressed particles over the entire hydrogen cycling procedure, we note that a corresponding $P_{abs}$ reduction for defect-free single crystals according to the SKG-model corresponds to a particle size decrease of $\sim 60$ % from a characteristic size of 500 nm to 200 nm[10]. For accuracy, it should be noted that the SKG-model, where $P_{abs}$ is intrinsically correlated to the strain state of the Pd system, strictly applies only to defect-free systems[9,10]. However, particles fabricated in a very similar manner as in this work have been shown to follow the SKG-model, provided they lack major grain-boundaries[11]. This justifies the approximate use of the SKG-model for the non-compressed particles in our system. Finally, we note that dislocations generated during nanocompression may serve as nucleation sites for the β-phase hydride, thereby lowering the thermodynamic barrier for hydride nucleation[59].

As a final interesting observation, we note that the evolution of $P_{abs}$ can be divided into five cyclic phases. They are clearly distinguished by plotting $P_{abs}$ vs. time spent in the hydride state (Fig. 3c): (I) the initial cycles with a decreasing absorption pressure for all particle groups; (II) cycle 20-40 with steady increase in $P_{abs}$ for all particle groups; (III) $P_{abs}$ again decreasing across the board; (IV-V) $P_{abs}$ again increasing, peaking and decreasing (more detail in SI section 12).

## Particle morphology analysis

As the next step, it is interesting to investigate particle morphology after hydrogen cycling since morphological changes provide further insights into the mechanism behind the evolution of $t_{30-50}$ and $P_{abs}$. Scanning electron microscopy (SEM) micrographs of cycled particles (SI Fig. 1) reveal that all particle types develop distinct protrusions on their surfaces during the first $\sim 20$ cycles (phase I in Fig. 3b, c). The protrusions appear to gradually increase in size, and they preferentially (but not exclusively) develop at or close to grain boundaries, if such boundaries are present (Fig. 4a–c and SI Fig. 1, 18). Here we should also add that protrusions that appear to grow at an angle to grain boundaries, i.e., not along the boundary, could stem from internal dislocations that terminated at the grain boundary, and that these dislocations then acted as fast diffusion paths to the surface that yielded the protrusion. Moreover, the grain boundaries in the particles may have migrated under the action of hydrogenation-induced internal stresses. The protrusion that forms at the original boundary position is then getting disconnected from the final position of a migrated grain boundary. This initial phase of the $H_2$ cycling, where the protrusions first appear, corresponds to the same phase in which the deceleration of the absorption kinetics sets in (cycle 0-20 in Fig. 3a). It also coincides with a decrease in $P_{abs}$ (phase I in Fig. 3b, c, for the non-compressed particles also confirmed on a second sample, SI Fig. 20).

The SKG-model, where $P_{abs}$ is intrinsically correlated to the strain state of the Pd system, thus implies that the appearance of the protrusions on the (non-compressed) particles is a stress-relief mechanism.

To further analyze this hypothesis, it is relevant to discuss the nature of the substrate supporting the particles, i.e., sapphire, which is a very hard single crystal. The reported value of adhesion energy between Pd and sapphire of 1.6 J/m² is significantly higher than between the other noble metals and sapphire[60]. This high adhesion energy, together with the high elastic modulus and hardness of sapphire thus implies strong clamping of the Pd nanoparticles to the substrate[60–62]. Also, we note that we neither ever have observed $\sim 50$ mbar $P_{abs}$ at 303 K (Fig. 3b, c) in our earlier works on hydrogenation of Pd nanoparticles[6,11] (high $P_{abs}$ implies higher strain levels according to the SKG-model), nor have seen protrusion formation on nominally identical Pd nanoparticles fabricated on softer Si or glass/fused silica substrates[11,12]. Accepting this fact then implies that the nominal volume expansion of up to 10% during Pd hydrogenation[63] combined with the strong adhesion to the sapphire substrate likely leads to significant clamping stress[19]. Together with the increased self-diffusion coefficient of Pd in the hydrogenated Pd lattice[64] which enables faster diffusion of Pd atoms from the highly strained interfacial region towards the less strained surface region of the particles, this high clamping stress at the Pd particle-sapphire interface explains the formation of the protrusions observed in the experiment.

To further corroborate these intermediate conclusions, we have prepared thin single particle cross-section lamellae of a 30-nm-compressed (Fig. 4d–f, and SI Fig. 21) and an as-fabricated particle (Fig. 4g–i, and SI Fig. 22). We selected this specific (as-fabricated) particle as it is a good representation of the average single particle behavior in terms of sorption kinetics and $P_{abs}$ evolution in this particle population. We used scanning transmission electron microscopy (STEM) and energy-dispersive X-ray spectroscopy (EDX) to analyze the lamellae of both particles after 76 hydrogenation cycles. Using EDX, this analysis corroborates that the protrusions indeed are made of pure Pd (Fig. 4f, i). The bright field (BF) images furthermore confirm the presence of defects inside the crystals, revealed as networks of darker contrast features in the BF images (Fig. 4d, g). The (dark contrast) defect features in the High-Angle Annular Dark-Field (HAADF) images confirm that many of these defects are likely associated with local voids or low-density regions inside the particles (Fig. 4e, h). Furthermore, a zoom-in of the non-compressed particle reveals extensive damage to the sapphire substrate at the interface (Fig. 4g), which is absent both in the compressed particle (Fig. 4d) and the non-compressed, non-hydrogenated one (Fig. 1b). This substrate damage is also reflected in traces of Al found in the magnified EDX map of the substrate-particle interface of the non-compressed particle (Fig. 4i). Such traces are absent in the EDX map of the compressed particle (Fig. 4f). Since sapphire has a shear modulus of $G \sim 150$ GPa, both findings imply high stress levels at the particle-substrate interface, specifically in the non-compressed particles, where initial stresses have not been released by plastic deformation upon nanocompression. An estimation of the stress levels induced in the particles upon hydrogenation yields values of the order of $\sim 1$ GPa (see SI section 18). Notably, these values are indeed comparable to the stress levels reached during the hydrogenation of annealed and nano-crystalline Pd thin films, as reported by Delmelle et al[46]., where this level of stress was sufficient to activate dislocation-mediated plasticity in both annealed and non-annealed samples.

As a further important aspect, we note that in both particle types a void has formed at the particle-substrate interface after hydrogen cycling (Fig. 4d, f and Fig. 4g, i). Such a void is absent in the non-hydrogenated particle (Fig. 1b). This void formation is likely caused by the migration of Pd atoms from the initially highly stressed interface region towards the surface of the particles, where they form the observed protrusions. The observed preferential formation of

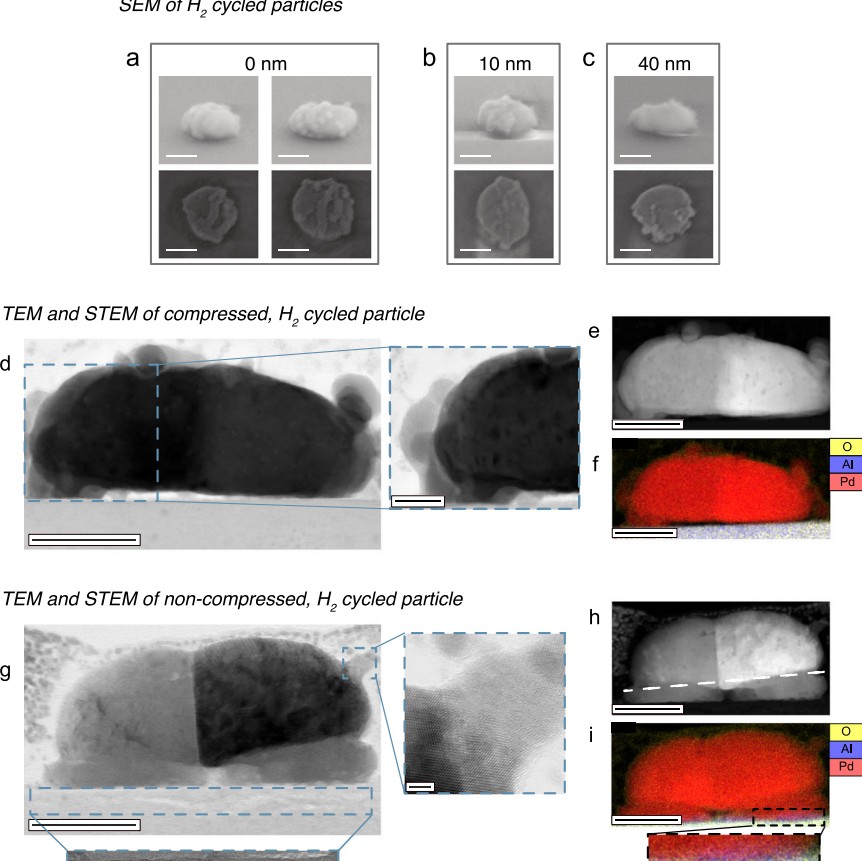

**Fig. 4 | Electron microscopy analysis of Pd particle morphology after 76 hydrogenation cycles. a–c** Side and top view high resolution scanning electron microcopy (SEM) images of a selection of particles with 0 (**a**), 10 (**b**) and 40 nm (**c**) nominal compression levels. Note the protrusions that have formed on each nanoparticle, irrespective of compression degree. Scale bars 100 nm. (**d**) Bright-field (BF) and (**e**) High-Angle Annular Dark-Field (HAADF) TEM micrographs together with an energy-dispersive X-ray spectroscopy (EDX) map (**f**) of a nominally 30 nm compressed nanoparticle. We note significant defect structures appearing as regions of dark contrast in both BF and HAADF micrographs. Dark contrast features in the HAADF micrographs are likely associated with local voids or low-density regions inside the particles. Scale bars 50 nm (**d–f**) and 20 nm (magnified area of d). The height of the particle is 70 nm. **g** Bright-field and (**h**) HAADF micrographs together with (**i**) an EDX map of a non-compressed nanoparticle. We note significant damage to the sapphire substrate as exemplified by the magnified HAADF

micrograph of the area outlined with a dashed box in (**g**), as well as the aluminum EDX signal extending up into the particle in the magnified area of (**i**), which indicates a significant roughness of the particle-sapphire interface. Since the shear modulus of sapphire is 150 GPa, and since this particle was not compressed, this implies that significant clamping stress has developed at the particle-sapphire interface during the $H_2$ cycling process. We also note from the EDX mapping in (f,i) that the particle protrusions consist of Pd only and with a different crystal orientation compared to the grain they are attached to, as exemplified by the zoom-in on a single protrusion in (**g**) (upper dashed box). Also, for this particle, note the presence of a significant number of defect structures revealed as regions of dark contrast in both BF and HAADF micrographs. The dark contrast features in the HAADF micrographs are likely associated with local voids or low-density regions inside the particles. Scale bars 50 nm (**g–i**) and 2 nm (magnified area of g). The height of the particle is 65 nm.

protrusions at grain boundaries intersecting the surface can be then attributed to the higher diffusion rate of Pd along grain boundaries compared to the bulk[65].

As the final aspect, we note that the non-compressed particle exhibits two distinct parts (Fig. 4h): a lower part that is more extended laterally along the substrate (Fig. 4h, below the dashed line), and a rounder upper part (Fig. 4h, above the dashed line). Most likely, these two parts are separated by grain boundaries. Notably, such a division into two parts is not seen in the compressed particle (Fig. 4e). Hence, together with the significant damage to the substrate for the non-compressed particle and the void formation, this suggests that the interfacial stress was high enough to plastically deform the interface region by both damaging the substrate and by separating the particle into two morphologically distinct parts, as also seen for other non-compressed particles (SI Fig. 1b). This separation resembles recrystallization–the process of formation of new grains in a heavily deformed metal. While bulk deformed Pd recrystallizes at the temperature of about 250 °C[66], high deformations at the Pd nanoparticle–

sapphire interface may shift the onset of recrystallization as low as to room temperature.

Taken together, the collected observations confirm the formation of Pd protrusions on the particle surfaces upon $H_2$ cycling, driven by high stress at the Pd-sapphire interface that induces a net diffusion of Pd away from that interface to release that stress. This is confirmed by void formation and facilitated by the enhanced Pd diffusivity in Pd hydride. The interfacial stress is highest for non-compressed particles that have not undergone plastic deformation prior to exposure to hydrogen, as corroborated by the sapphire substrate damage observed in this case, and the observed shearing of the particles into two parts.

As a second aspect, we can also conclude here that the observed differences in $t_{30-50}$ and $P_{abs}$ between compressed and non-compressed particles (cf. Figure 3) are not due to size variations, since these are too small (SI Fig. 18b). Specifically, after the full hydrogen cycling, the recrystallization processes discussed here result in convergence of sizes of the compressed and non-compressed

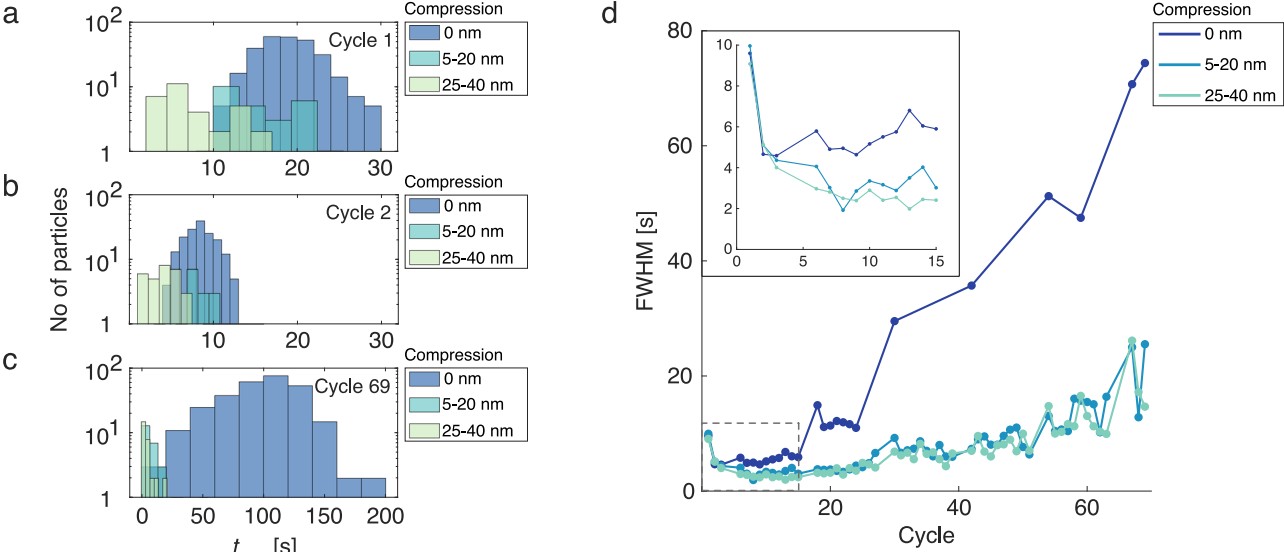

**Fig. 5 | Histograms of the single particle $t_{30\text{-}50}$ values.** Histograms for cycle 1 (**a**), 2 (**b**) and 69 (**c**). The particles are divided into three groups depending on their degree of compression, i.e., as-fabricated (0 nm compression), 5–20 nm compression and 25-40 nm compression. We note the significant decrease of the average $t_{30\text{-}50}$, as well as a substantial narrowing of the distributions from cycle 1 to cycle 2. This is followed by an extreme broadening of the distribution for the as-fabricated particles at later cycles, as exemplified by the histogram from cycle 69

(**c**) note the different scale on the x-axis). **d** The full width at half maximum (FWHM) of the $t_{30\text{-}50}$ absorption time histograms plotted for the as-fabricated (0 nm compression), the 5–20 nm and the 25–40 nm compression groups as function of cycle number were calculated from a normal distribution fitted to the $t_{30\text{-}50}$ histograms. For the FWHM evolution of all individual compression-level sub-groups, see SI Fig. 23. Inset: Magnification of the FWHM evolution for the first 15 cycles (outlined with a dashed box in the main figure).

particles, such that the difference in diameters even between the most compressed particles (40 nm) and their non-compressed counterparts is relatively small, i.e., less than 10% (SI Fig. 18b). This, combined with the observed hydrogenation behavior (i.e., the fact that the largest difference in kinetics is observed at the latest cycles (Fig. 3) without significant overall change in transition pressures) effectively rules out shape or geometric factors as explanations for the observed effects.

**Single particle statistics**

Finally, having established that the particles' hydrogenation kinetics slow down upon hydrogen cycling and that the particles undergo significant restructuring during the process, it is of interest to look closer at the single particle statistics behind the evolution of the absorption kinetics (Fig. 5, for the same analysis for $P_{abs}$, see SI Section 20, for more detailed single-particle kinetics and isothermal data, see SI Fig. 19 and SI Section 22). By fitting a normal distribution to the histograms of the single particle $t_{30\text{-}50}$ values for individual cycles (Fig. 5a–d), we see that not only the absorption kinetics for all particles slow down, but also that the width of the $t_{30\text{-}50}$ distribution increases simultaneously, i.e., $t_{30\text{-}50}$ of the particles become increasingly scattered the more time the particles have spent in the hydrogenated state. Again, this increase in variance is more pronounced for the non-compressed particles (which is also confirmed on a second sample, see SI Fig. 29f). For the compressed particles on the other hand, the increase in full width at half maximum (FWHM) of the $t_{30\text{-}50}$ distributions is less pronounced. This means that along with accelerating the hydrogen absorption kinetics, plastic deformation also acts as a stabilizer of the accelerated response over time.

One possible explanation for this absorption time stabilizing effect of nanocompression could be the reduced strain levels in the compressed particles, as inferred from the lower absorption plateau pressures (cf. Figure 3b). Hence, higher stress levels during hydrogenation of the non-compressed particles act as a driving force for these particles to develop features that accelerate kinetics, e.g., by releasing this stress by the formation of new (absorption accelerating) dislocations, as decided by the stress state of every individual particle.

To this end, it is well established that the nucleation of dislocations in pristine or dislocation-starved nanoparticles at the onset of plasticity is highly stochastic[37,67]. The level of internal stresses developing in the non-compressed particles during hydrogenation cycling process is comparable to their ultimate strength[21]. Therefore, we conjecture that the dislocation substructure in these particles develops in a highly stochastic manner, depending on their individual initial dislocation state and their (stochastic) dislocation nucleation history. The resulting variability of dislocation substructures between the individual particles may indeed explain the increase of $t_{30\text{-}50}$ spread upon cycling seen in Fig. 5d.

As a consequence, this means that the origin of the hydrogen absorption kinetics stabilizing effect of nanocompression is likely found in the dislocation networks generated during the plastic deformation and the fact that they preferentially form close to the particle surface where they can annihilate[26–29,34,35]. Since the rate-limiting step for hydrogen absorption into Pd is diffusion of dissociated hydrogen from the surface to the first sub-surface layer[51,68], we argue that the specific energetics of the dislocation exit points, which constitute sites of lower coordination than the adjacent terraces, are likely to enhance the overall absorption kinetics both by lowering the activation barrier and by providing access to "hydrogen diffusion highways", i.e., dislocation cores exhibiting higher diffusivity of hydrogen compared to the Pd lattice[12,46,47,69].

In conclusion, using in-situ nanocompression combined with STEM microstructure characterization and multiplexed plasmonic nanoimaging microscopy of large arrays of individually deformed Pd nanoparticles, we have studied the effect of systematic plastic deformation on hydrogen absorption kinetics at the individual nanoparticle level. These experiments have been correlated with optical pressure-composition isotherm measurements, from which the Pd α-to-β phase transition hydrogen pressure could be inferred. The obtained results show a clear compression-level dependent trend, where more heavily deformed particles exhibit both accelerated hydrogen absorption kinetics and lower α-to-β phase transition $H_2$ pressures when exposed to hydrogen gas. We identify dislocation formation during the

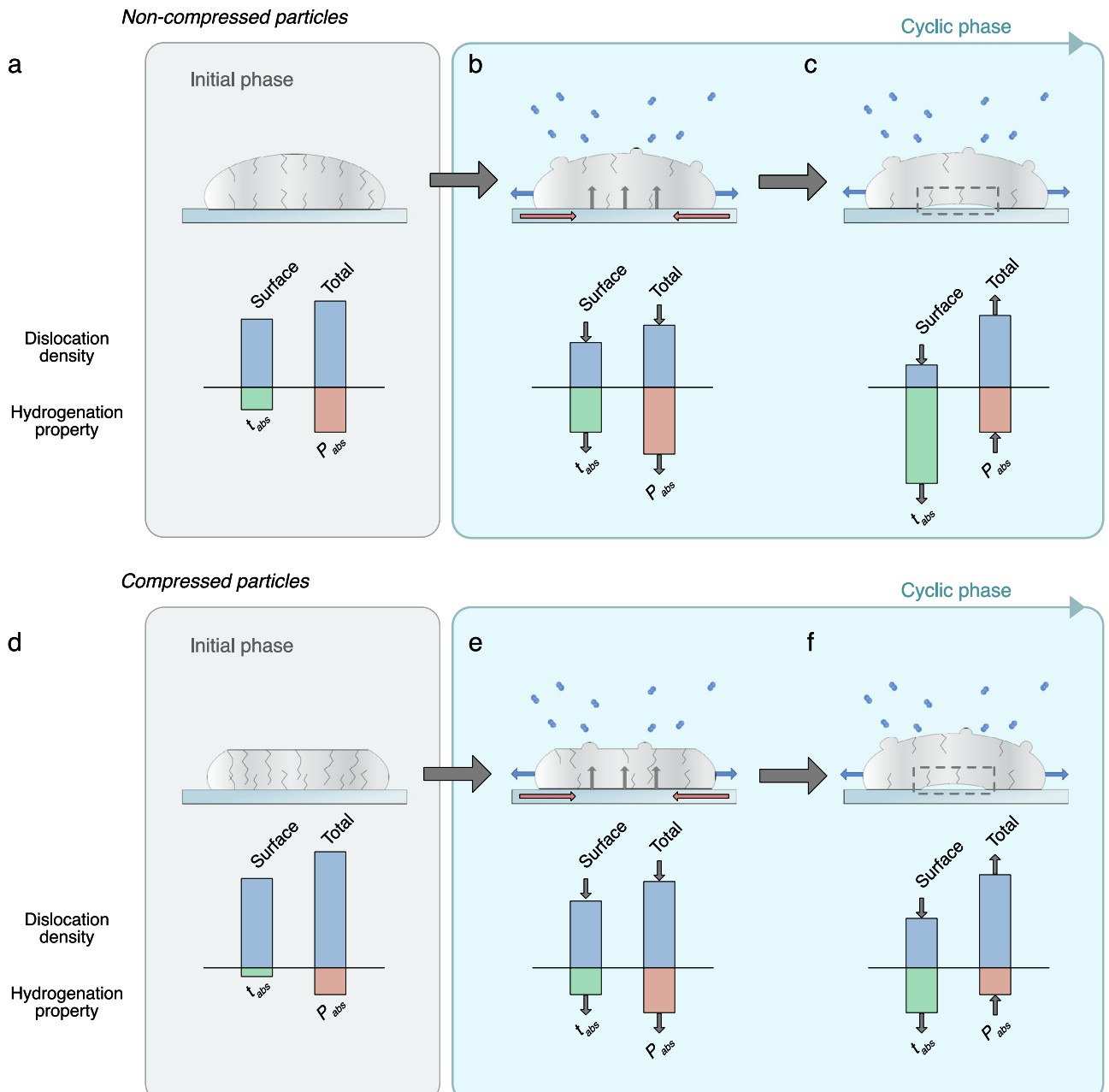

**Fig. 6 | Summary of particle morphology evolution and its impact on hydrogen absorption.** Schematic showing the morphology evolution of the non-compressed (**a**–**c**) and compressed (**d**–**f**) Pd particles upon hydrogen cycling together with the observed impact on hydrogenation properties. Initially, the non-compressed particles exhibit lower dislocation densities (**a**) compared to the compressed particles (**d**). This leads to slower hydrogen absorption kinetics ($t_{abs}$) and higher hydride phase transition pressures ($P_{abs}$) for the non-compressed particles. After multiple hydrogenation cycles (**b**, **e**), the dislocation densities in both particle types have decreased by annihilation. As a consequence, hydrogen absorption has slowed down and the phase transition pressures have increased for both particle types. At this stage, the annihilation of dislocations is mainly mediated by stresses generated by the nucleating and growing hydride phase, whereas the high interfacial stress due to substrate clamping leads to net Pd diffusion to the free surface region, preferably along grain boundaries, where Pd protrusions gradually form. After further cycling (**c**, **f**), dislocations at the free surface continue to annihilate, leading to even slower kinetics ($t_{abs}$). In the core and substrate interface region of the particle (dashed box), however, the low density of dislocations that terminated the last phase of the cycling (**b**, **e**) lead to high stresses during hydrogenation. Consequently, when reaching a threshold level, the particles relieve this stress by forming new dislocations in the core and substrate interface region, leading again to a decrease of phase transition pressures $P_{abs}$. Subsequently, the particles continuously repeat the dislocation formation/annihilation cycle in the core and substrate interface regions, all while dislocations at the surface continuously annihilate (**b**, **c** and **e**, **f** outlined with light blue box). This thus leads to a cyclic trend of $P_{abs}$ and monotonously decreasing absorption kinetics ($t_{abs}$). The compressed particles have consistently both faster kinetics ($t_{abs}$) and lower phase transition pressures $P_{abs}$ over the entire studied time frame due to their initial compression.

nanocompression as the main mechanistic reason for both effects. The dislocations serve as diffusion-accelerating pathways enhancing hydrogenation kinetics and contribute to the reduced $P_{abs}$, either directly by acting as nucleation sites for the β-phase hydride, or indirectly by lowering the strain levels of the particles (Fig. 6a, d), or both.

As a second aspect, we have studied how the hydrogen absorption kinetics and the α-to-β phase transition pressures of the individual particles evolve upon repeated cycling with hydrogen up to 76 cycles. Our findings reveal that, after an initial acceleration, kinetics decelerate upon repeated cycling and the particles show a high variability in

the time evolution of their absorption kinetics. This effect is most pronounced for the non-compressed particles whose average absorption time increases from 6 s to 109 s. We attribute this slow-down of the kinetics to a continuous annihilation of dislocations during the hydrogen cycling[12], particularly those near the particle surface, since the diffusion of hydrogen from surface to sub-surface sites is the rate limiting step in the overall absorption process[51].

The repeated hydrogenation, on the other hand, results in cyclic variations of the absorption phase transition pressures, $P_{abs}$, where $P_{abs}$ initially increases, followed by a subsequent decrease, all while the particles slowly become more and more identical in terms of $P_{abs}$ (SI section 12, 20 and 23). We postulate that this cyclic trend of $P_{abs}$ implies a cyclic formation and annihilation of dislocations in the particles (Fig. 6), most pronounced at the highly strained particle-substrate interface, as has been reported for thin Pd films[62]. This could explain the current discourse in the literature on whether hydrogen cycling increases or decreases the dislocation density in Pd nanoparticles[12,13,21,23], where the earlier studies had tracked the morphological evolution of the particles over much shorter timescales than those employed in the present work, and as such would not have captured this cyclic behavior.

In summary, our findings demonstrate the importance of morphology and dislocations for the hydrogen absorption in Pd nanoparticles, and more specifically, the ability of plastic deformation to completely change the initial properties and evolution of the hydrogen absorption process. These results are highly relevant for nanostructured metal hydrides that find application in hydrogen storage and for nanoparticle-based hydrogen sensors, as the modification of absorption and desorption pressures, as well as fast and stable kinetics over many hydrogenation cycles, are of utmost importance. Looking forward, fully mapping the formation and annihilation dynamics of dislocations during hydrogen sorption will require a measurement technique, or a combination thereof, that i) can capture the full dislocation network, both at the core and the surface level, of multiple single Pd nanoparticles at ii) long enough timescales to fully capture the potential cyclic annihilation/formation of dislocations during the cycling while simultaneously iii) capturing the hydrogenation kinetics of the process for all involved particles. Combining Bragg coherent diffractive imaging (BCDI) to capture the dislocation networks of the particles[13,23] with intensity based single particle plasmonic nanoimaging to follow the hydrogenation kinetics[12,70] could constitute such a solution.

## Methods
### Sample preparation
The sample was prepared on $c$-plane oriented, double-polished sapphire wafer of 500 μm in thickness and 4 inches in diameter. To fabricate the particles by means of nanolithography[71], we used electron-beam lithography (JEOL JBX 9300FS) to define the position and diameter of the particles by patterning a polymer resist (MMA 8.5 MMA-EL6 2000-1000-1 min, HP 180 C 5 min, PMMA 950-A2 2200-1100-1 min, HP 180 C 5 min; development: 3:1 MIBK:IPA 1 min, rinse in IPA, blow dry), and subsequent Pd evaporation through the predefined mask with 5 s of $O_2$ plasma cleaning (50 W, dry etch RIE, Plasma-Therm) before metallization. Metal deposition was performed using an e-beam thin film evaporator (Kurt J. Lesker PVD 225) with a base pressure of $3 \times 10^{-8}$ Torr. Lift-off of the mask occurred overnight in Microposit™ Remover 1165. After fabrication, the samples were annealed at 500 °C for 2 h in a tube furnace (Nabertherm R50/250/12) under the flow of Ar-2 vol.% $H_2$ gas (300 mL min⁻¹).

### Nanocompression
The in-situ nanocompression was performed with a Hysitron PI85 PicoIndenter fitted with a flat diamond square punch of $1 \times 1$ μm² in size, which uses a three-plate capacitive design to apply load and measure displacement simultaneously. The PicoIndenter was fitted into Zeiss Ultra-Plus SEM. The nanocompression was carried out as follows: the tip was slowly brought into contact with the particle (indicated by a small but distinct increase in load). It was then further lowered by an additional $x$ nm, where $x$ corresponds to deformation levels of 5, 10, 15, 20, 25, 35 and 40 nm.

### Single particle kinetics measurements
The sample was placed in a vacuum-tight microscope chamber (custom-made Linkam 350 V) and secured with Kapton tape (HB830, Hi-Bond Tapes Ltd) to eliminate movement during the measurement. To enable back-scattering of light through the transparent sapphire substrate, a reflecting piece of Si was fixed underneath the sample. The chamber has a heating stage that keeps the sample at 303 K during the entire measurement. The chamber outlet of the stage was connected to a vacuum pump (Pfeiffer, TSU 071) via a solenoid valve (custom modified Peter Paul 2209DGM) and evacuated to ~1 μbar pressure. To make sure that a $H_2$ pulse was fast, mass flow controllers (Bronkhorst) were used to build up 100% $H_2$ gas (6.0 purity) in the tubes leading to the closed chamber prior to measurements. To then measure the kinetics of the Pd nanoparticles, the image of the nanoparticle array was continuously captured with a thermoelectrically cooled electron multiplying charge coupled device (EMCCD) camera (Andor iXon Ultra 888) with a scan rate for image acquisition of 2 frames per second through an upright optical microscope (Nikon Eclipse LV100, Nikon 50× BD objective) equipped with a motorized stage (Märzhäuser). The camera, valves and mass flow controllers were controlled simultaneously by a custom LabVIEW program. The sample was initially measured for 5 min in vacuum before the gas inlet valve (custom modified Peter Paul 2209DGM) was opened to let in a ~350 mbar pulse of $H_2$. The system was then allowed to saturate for 5 min (10 min after cycle 54 and onward), after which the pump valve was opened to pump the chamber back to the initial ~1 μbar pressure for 20 min (cycle 1-3), 40 min (cycle 6-51) and ~ 3 h (cycle 67-70). The reason for the increased time for hydrogenation saturation and pumping was the deacceleration of the particles' kinetics for later cycles. For more specific information about the hydrogenation parameters for every individual cycle, see Supplementary Data 1. For every frame, the intensity for each particle was then calculated as the sum of the brightest pixel and its four nearest neighbor pixels in the diffraction-limited spot of light from that particle. The intensity change of the particle during hydrogen absorption and desorption is proportional to the hydrogen concentration in the Pd-containing particle[11,12,43].

### Single particle isotherm measurements
The experimental setup and process were the same as for the kinetics measurements, except that the measurement chamber outlet now was connected to an (atmospheric pressure) exhaust instead of a vacuum pump, which allows for continuous flow experiments at atmospheric pressure. During isotherm measurements, the concentration (and thereby the partial pressure) of $H_2$ in Ar carrier gas was slowly increased from 10 to 100 mbar, with at least 300 s per concentration step for a total time of 21 300 s during absorption measurements. For desorption, respectively, the $H_2$ concentration was initially set to 60 mbar (100 mbar after cycle 28) for 15 min (40 min for cycle 76) to hydrogenate the particles, after which the $H_2$ concentration was dropped to ~ 30 mbar and thereafter slowly decreased down to 1 mbar with 300 s per concentration step (2400 s per step for cycle 76). For more specific information about the hydrogenation parameters for every individual cycle, see Supplementary Data 1. All measurements were taken at 303 K.

## SEM imaging

SEM imaging was performed in a Zeiss GeminiSEM-450 at an acceleration voltage of 0.5 kV and low-current conditions ($\sim$ 20 pA beam current).

## FIB lift-out and TEM imaging

The cross-sectional lamellas used for TEM imaging was prepared by the lift-out method in a dual-beam FIB (FIB; FEI Helios NanoLab Dual-Beam G3 UC). The subsequent TEM imaging was performed in a FEI Themis G2 300 80–300 keV S/TEM. EDX mappings were performed using a Dual-X detector (Bruker). Camera length was 94 mm at 300 kV.

## Data availability

Source data acquired and generated in this study have been deposited in the Zenodo digital repository under DOI code 15553768[72].

## Code availability

The MATLAB code generated and used in this work is available at: (https://gitlab.com/langhammerlab/hydrogenation-of-compressed-nanoparticles/-/tree/a048ee1d71e2cdd9946fd1c8d63e5ce7e15cce64/) and under "Software" in the Zenodo repository[72].

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

## Acknowledgements

This research has received funding from the Knut and Alice Wallenberg Foundation project KAW 2018.0459 (CL), the Swedish Foundation for Strategic Research project SIP21-0032 (CL) and Israel Science Foundation, grant No. 617/19 (ER). Part of this work was carried out at the Chalmers MC2 cleanroom facility and at the Chalmers Materials Analysis Laboratory (CMAL) and under the umbrella of the Chalmers Nano Area of Advance.

## Author contributions

The manuscript was written through contributions of all authors (C.A., J.Z., J.F., E.R., C.L.). All authors have given approval to the final version of the manuscript.

## Funding

## Competing interests

The authors declare no competing interests.
