## [Transparent Peer Review file · Nature Communications]

Hydride Formation Pressures and Kinetics in Individual Pd Nanoparticles with Systematically Varied Levels of Plastic Deformation

Corresponding Author: Professor Christoph Langhammer

Version 0:

Reviewer comments:

Reviewer #1

(Remarks to the Author)

The paper describes a detailed study about the kinetics of H₂ sorption into Pd nanoparticles (NPs) predeformed up to different levels. High pre-deformation increases significantly the rate and amount of H₂ that can be stored and this effect persists after many cycles. The reason for this effect is mainly connected to the defects in the NPs.

The main merits of this work are the quality and quantity of challenging experiments. Compressing NPs at different levels of deformation, followed by H₂ sorption and desorption cycles with in situ plasmonic nanoimaging to estimate absorbed H₂ from scattering intensity is really a major achievement. The study is supplemented by TEM cross-section analysis. There is no doubt that the proposed series of experiments is pushing the frontier of the field and gives access to novel data.

However, the interpretation and use made of these data remains qualitative and, from there, the discussion remain relatively speculative. The study lacks an effort to put numbers and equations to validate or not some of the proposed scenarios. This includes the following aspects:

1. A quantitative mechanical analysis of the state of stress and strain in the NP is almost totally absent. We refer here to :
 - 1.1. the state of stress after film deposition – what is the residual stress level ? (maybe from curvature measurements), then the state of residual stress after NP patterning (which will lead to some degree of the stress release)
 - 1.2. the state of stress after annealing (more difficult to evaluate)
 - 1.3. the state of stress after NP compression. This is really where quantitative information is missing and should be generated. The authors report only total displacement imposed to the NP. But this should be converted into global strain levels (as well as global stress from compression data, even if approximate), either using approximate uniaxial compression model or using FE simulations (to better take into account the constraint at the substrate NP interface). The global strain is the sum of elastic strain and plastic strain. What matters here is the amount of plastic strain that has been imposed and which is directly reflected into a change of dislocation density. To calculate elastic strains, one needs an approximate yield stress value of the Pd NPs. Then from FE simulations, one can extract local stress and strain gradients that will definitely build up due to the nature of the specimens (relatively Nps, attached to substrate). Obviously, solving the problem accurately is difficult as one needs a complex crystal plasticity model, but it is better to do something than nothing based on simple plasticity model.
 - 1.4 the state of stress with H₂ for which, again, different levels of modelling can be attempted, from simple homogenous imposition of a dilatational eigenstrain (e.g. fictitious thermal expansion coefficient) in a closed form model to more complex FE simulations. The constraint by the substrate is indeed the key in the change of residual stresses. This would give an idea of the magnitude of the stress that can be attained during H₂ loading. There are several papers in the literature explaining how to convert H₂ fraction (as a function of the phase) into stress, for instance in works on thin Pd films with H₂ loading.
2. Modelling the diffusion process and associated energetics. Here as well, the authors could have a look at relatively simple

(though approximate) models, maybe only to compare extreme cases. One can make assumptions on the diffusion coefficients associated to piping diffusion in dislocations and GB, from literature, and justify the ratio of kinetics with or without dislocations.

3. An interesting observation is that specimens with a small dislocation density initially (such as uncompressed NPs) may, under H₂ sorption, lead to a starvation of the dislocations ("starvation" is the word that is used in the dislocation community to express the "cleaning" of a material that get rid off dislocations by mechanical stress). While with a sufficiently high dislocation density, dislocation multiplication mechanisms will start playing a role. I remember papers in Nature Communication or Nature Materials simulating with discrete dislocation dynamics compression of nanopillars and looking at these phenomena. They provide data out of which one can possibly define a threshold dislocation density separating the two regimes, as a function of specimen size.

This is not to say that the authors should pursue all these lines, but the value and impact of the work would increase a lot if some of the experimental findings could be put into equations.

Finally, the analysis of the defects needs to go further. The authors are invited to consult the literature on the subject for instance a series of paper by Idrissi and coworkers on defects in Pd films which revealed numerous features including, on top of dislocations, twins, grain growth, production of five folds twins, 9R phase, stacking faults which vary in terms of deposition conditions as well as H₂ sorption levels. In other words, the initial defect structure can be more complex than "only dislocations" and can evolve in a complex manner as well under the very large stress that can be built during H₂ sorption (showing again the need to quantify a little more the magnitude of these stresses).

Reviewer #2

(Remarks to the Author)

The paper contains an extensive analysis of the absorption-desorption behavior of over 300 single Pd nanoparticles (deformed and undeformed) on a sapphire substrate with adequate statistical support of the findings.

The result is a very comprehensive understanding of the processes involved in absorption and desorption of hydrogen in Pd i.e. from the clamping effect to the formation and annihilation of dislocations and development of strain. Although these phenomena are described earlier this paper shows clear evidence. Most interesting are the visualizations of the changes in morphology /shape/protrusions resulting from H-induced deformation mechanisms. The discussion is underpinned by data, the complete data sets in the Supplementary allow verification although the color intensity differences in the figures are sometimes hard to distinguish.

Comparing the undeformed and deformed particles raises questions at some points. Can the authors argument on these to support their conclusions.

(1) Surface contamination by oxides at fabrication - deformation could damage the oxide layer and enhance H uptake
(2) op p 12/13 ref 9 and 46 are mentioned to indicate that compression could lower the coherency strain induce hydrogenation barrier - the relation to compressive deformation is unclear. And: The strains in that framework are proportional to the systems volume - and discussed for macroscopic systems - in the case of nanoparticles the surface plays an important role and allows faster relaxation and dislocation annihilation - I doubt if the model is fully applicable to nano-sized systems.

p.15 Clamping effect: the discussion is about the softness of the substrate but how about the chemical interaction with the substrate - the bonding between the particle and substrate plays a major role.

Clamping of the undeformed particles is mentioned to be stronger, why? The surface area of deformed particles in contact with the substrate is expected to be larger - what is the influence of that on the sorption behavior?

On p22: discussion on the slow down of the H sorption kinetics - the undeformed particles showing the largest slow down; in the paper this is attributed to dislocation annihilation (indeed could be a reason) unclear to me why in the deformed particles the annihilation would be less. Is there any proof for that statement. After many cycles it is expected that the differences in dislocation densities between both type of particles reduces.

After clarification and discussion of the above points raised, the study is eligible for publication.

Reviewer #3

(Remarks to the Author)

In this manuscript, the authors study the kinetics and the structural consequences of hydrogen absorption in Pd nanoparticles that have previously undergone different levels of plastic deformation. The manuscript is of high-quality, very dense, with significant results and is well suited for publication in Nature Communications.

I agree with most of the conclusions drawn by the authors. However, I would appreciate if the authors could consider the following comments:

- The discussion regarding the influence of dislocation on hydrogen absorption is based on the measurement of the average t_{30-50} absorption times (Figure 3, Supplementary figure 6), the plateau pressures (Supplementary figure 10). In these figures, the error bars represent the standard error. The variations of t_{30-50} and the plateau pressures vs cycle cannot be considered as significant as all error bars overlap (except for the evolution of t_{30-50} for the uncompressed particles compared with that for the compressed ones). I think the authors should include a statistical analysis to support their conclusions.

- Line 269: "existing dislocations can reach the surface and shear the crystal upon hydrogenation". Is the behavior of monocrystalline nanoparticles different from that of polycrystalline ones, which contain not only dislocations but also grain boundaries?

- Figure 4d-g: the authors conclude from the bright-field TEM images that "significant dislocation structures appear(...) as regions of dark contrast in both BF and HAADF micrographs". It is also said that "Dark contrast features in the HAADF micrographs can also be interpreted as local voids or low-density regions inside the particles".

o The camera length used for HAADF imaging is not given, so it is difficult to evaluate whether dislocations can be observed using this imaging mode.

o If we consider that the images reveal mass-density contrasts as expected in HAADF, the features should rather be voids or low-density regions. As the contrasts in the BF images are similar to those observed in HAADF, it is difficult to draw a clear conclusion about dislocation

As minor comments:

- Supplementary figure 1: the SEM images of the nanoparticles are very small. It is very difficult to analyze in detail whether the particles are considered as single crystalline or polycrystalline. Moreover, the particles are numbered in Supplementary figure 2, but the numbers are not given in Supplementary figure 1. Please note that a few scale bars are also missing.

- Figure 1b: a few dislocations are indicated by white and red arrows. It is difficult to distinguish them from all other contrast variations within the particle. In order to better see the dislocations, could the authors provide Weak Beam Dark Field images (if they used this imaging mode)? Moreover, could they provide an indexation of the highlighted dislocations?

- Figure 1j-k: in the literature, monocrystalline nanoparticles have already shown stepwise-staircase curves. Why do the authors say that such curve is characteristic of polycrystalline particles?

- Figure 2f: why is the error bar far larger for uncompressed particles than compressed ones?

- Line 279: "Importantly, such sessile dislocation structures can survive the process of hydrogen cycling". Why? Does a reference support this assertion?

- Line 401: "This void formation is likely caused by the migration of Pd atoms (...) towards the surface of the particles, where they form the observed protrusions". Why are the protrusions not close to grain boundaries in polycrystalline particles?

- Line 242: I think "size" would be more appropriate than "shape".

- Line 428-430: "Upon compression, the particle shape becomes more disk-like... on hydrogenation kinetics". I do not see the link between this sentence and the previous ones.

- Lines 450-452: "As the nucleation of dislocations in the non-compressed particles is highly stochastic...". Please explain, this sentence is unclear.

- Several figures from the Supplementary file are not discussed in the manuscript. For instance, I did not understand what they bring (for instance the graphs showing PH₂ in function of the average dark-field scattering intensity). Also in Supplementary figures 20 and 21, linear fits are given. In several cases, a statistical analysis (linear regression) may indicate that the slopes are not significantly different from 0. Please check.

K. Masenelli-Varlot

Version 1:

Reviewer comments:

Reviewer #1

(Remarks to the Author)

The authors have provided convincing answers to my (challenging for several) suggestions and queries. Most of my comments aimed at pushing the quantitative-mechanical-modelling aspects as far as possible. I did and do realize that the

complex nature of the system with a lot of variations from particles to particles preclude quantitative predictive modelling. I'm certainly happy with the new efforts put to provide bounds and/or indicate along which line the work should be pursued to build physics based models. These answers also convince me about the expertise of the authors around these questions of residual stresses, plasticity, dislocation mechanics and of their good knowledge of the literature. I propose thus to accept the paper in view of the excellent and very novel set of data extracted from this difficult investigation.

Reviewer #2

(Remarks to the Author)

The authors provided the necessary information, explanation as an answer to the questions raised in the referee report. The reply is underpinned with appropriate argumentation and proof. The paper is ready for publication in the revised form.

Reviewer #3

(Remarks to the Author)

The authors answered all the comments expressed in the first review in a very convincing way. Both the manuscript and the supporting information file have been greatly improved. I have no other question and the revised manuscript deserves to be published as it is.

Reviewer #1 (Remarks to the Author):

The paper describes a detailed study about the kinetics of H₂ sorption into Pd nanoparticles (NPs) predeformed up to different levels. High pre-deformation increases significantly the rate and amount of H₂ that can be stored and this effect persists after many cycles. The reason for this effect is mainly connected to the defects in the NPs.

The main merits of this work are the quality and quantity of challenging experiments. Compressing NPs at different levels of deformation, followed by H₂ sorption and desorption cycles with in situ plasmonic nanoimaging to estimate absorbed H₂ from scattering intensity is really a major achievement. The study is supplemented by TEM cross-section analysis. There is no doubt that the proposed series of experiments is pushing the frontier of the field and gives access to novel data.

Our reply: We thank the Reviewer for this overall positive assessment of our work.

However, the interpretation and use made of these data remains qualitative and, from there, the discussion remain relatively speculative. The study lacks an effort to put numbers and equations to validate or not some of the proposed scenarios. This includes the following aspects:

1. A quantitative mechanical analysis of the state of stress and strain in the NP is almost totally absent. We refer here to :

1.1. the state of stress after film deposition – what is the residual stress level? (maybe from curvature measurements), then the state of residual stress after NP patterning (which will lead to some degree of the stress release)

Our reply: Even though we agree with the Reviewer that approximating the strain of the particles after fabrication would be interesting (see next answer below as well), we must here clarify that our fabrication technique does not involve film deposition followed by the nanoparticle patterning. In fact, it is precisely the other way around as we first nanofabricate a mask for the particles using electron-beam lithography, and then deposit the metal directly through the holes in the mask to create the nanoparticle disks. In other words, we never form a continuous thin film. As an example to illustrate this kind of nanolithography, we refer to the paper “Fredriksson, H. et al. Hole-mask colloidal lithography. *Advanced Materials* **19**, 4297-4302 (2007). <https://doi.org/10.1002/adma.200700680>” where self-assembly is used for patterning of the mask rather than electron-beam lithography as we use in the present work. To make the used nanofabrication method clearer, we have updated the Methods section of the main text as:

“To nanofabricate the particles by means of nanolithography⁷⁴, we used electron-beam lithography (JEOL JBX 9300FS) to define the position and diameter of the particles by patterning a polymer resist (MMA 8.5 MMA-EL6 2000-1000-1 min, HP 180C 5 min, PMMA 950-A2 2200-1100-1 min, HP 180C 5 min; development: 3:1 MIBK:IPA 1 min, rinse in IPA, blow dry), and subsequent Pd evaporation through the predefined mask with 5 s of O₂ plasma cleaning (50 W, dry etch RIE, Plasma-Therm) before metallization.”

As a second point, we would argue that the strain state of the particles directly after deposition of the Pd into the masks is less important than the strain state after annealing, which we discuss in the comment below, since the recrystallization induced by the annealing essentially “erases” both initial microstructure and stress state.

1.2. the state of stress after annealing (more difficult to evaluate)

Our reply: At the onset of annealing all nanoparticles (NPs) are polycrystalline, as they inherit the microstructure of as-deposited islands. Under these conditions, all deposition stresses in the islands should relax upon heating to the annealing temperature of 500 °C. The mechanism of this relaxation is a wedge-like grain boundary self-diffusion¹. A simple estimate based on the cited work of Gao et al. has shown that during fast heating of thin polycrystalline Ni films all biaxial stresses in the film relax between 250 and 300 °C². As Ni and Pd have close melting points, one can expect that all internal stresses in Pd islands relax at the heat treatment temperature of 500 °C employed in the present work. Upon cooling, the biaxial tensile stress develops in NPs due to the mismatch of coefficients of thermal expansion (CTEs) of Pd and sapphire. The upper bound of this stress can be estimated assuming that no stress relaxation occurs in the particles upon cooling. This is most likely true for the single crystalline NPs (like the one shown in Fig. 1(c) of the main text). Some stress relaxation certainly occurs in oligocrystalline NPs (like the ones shown in Fig. 1(d-e) of the main text), but the degree of stress relaxation depends on the geometry of grain boundaries, cooling rate, boundary conditions at the NP-substrate interface, etc..

Moreover, contrary to the thin films firmly attached to the substrate, the thermal stress in the NPs is highest at the interface with the substrate and decays exponentially towards the upper facet of the particle. Therefore, we will only provide the upper limit of biaxial tensile stress in the particles close to the substrate, σ , assuming no stress relaxation and no sliding condition at the interface:

$$\sigma = \Delta\alpha \times \Delta T \times Y \quad (1)$$

where $\Delta\alpha$, ΔT and Y are the difference of the linear thermal expansion coefficients of Pd ($12.85 \times 10^{-6} \text{ 1/K}$)³ and sapphire ($7.5 \times 10^{-6} \text{ 1/K}$)⁴, the difference between the annealing and room temperatures ($\Delta T = 480 \text{ }^\circ\text{C}$), and the biaxial elastic modulus of Pd, respectively. We will approximate the latter by $E/(1-\nu)$, where E is the isotropic Young modulus of Pd (121 GPa), and $\nu \approx 0.39$ is the Poisson's ratio⁵. With these data, $\sigma \approx 480 \text{ MPa}$, which is below the dislocation nucleation stress in pristine single crystalline Pd NPs⁶, but may be above the nucleation stress in oligocrystalline NPs, or in single crystalline NPs with high dislocation density. Therefore, we can conclude that Pd NPs after annealing are under biaxial tensile stress, with the upper limit of near-interface stress of about 500 MPa in single crystalline NPs. The stress levels in oligocrystalline NPs and in the NPs containing dislocations may be somewhat lower.

1.3. the state of stress after NP compression. This is really where quantitative information is missing and should be generated. The authors report only total displacement imposed to the NP. But this should be converted into global strain levels (as well as global stress from compression data, even if approximate), either using approximate uniaxial compression model or using FE simulations (to better take into account the constraint at the substrate NP interface). The global strain is the sum of elastic strain and plastic strain. What matters here is the amount of plastic strain that has been imposed and which is directly reflected into a change of dislocation density. To calculate elastic strains, one needs an approximate yield stress value of the Pd NPs. Then from FE simulations, one can extract local stress and strain gradients that will definitely build up due to the nature of the specimens (relatively Nps, attached to substrate). Obviously, solving the problem accurately is difficult as one needs a complex crystal plasticity model, but it is better to do something than nothing based on simple plasticity model.

Our reply: All Pd NPs tested in our work had approximately the same height of 60 nm, since they were fabricated simultaneously by e-beam physical vapor deposition. The total displacements given in our work refer to the load-displacement curves obtained in in-situ NPs compression (Fig. 1(j-k) of the main text). The displacement is defined as residual displacement at the bottom of the unloading branch of the curve. The slope of this unloading branch is determined by the relaxation of compressive elastic stresses in the particle and, hence, the displacement determined from the load-displacement curves provides a true difference of heights of as-fabricated and compressed NPs, after the load has been released. Therefore, the displacements given in our manuscript (in nm) can be easily translated into total plastic compressive strains just by dividing them by the initial particle height (of 60 nm): 0.083, 0.167, 0.250, 0.333, 0.417, 0.500, 0.583, and 0.667.

Unfortunately, constitutive plasticity models employed in FE simulations are optimized for bulk materials and are unsuitable for NPs with their high values of strength (in the GPa-range⁶) and plasticity modes depending on dislocation nucleation and their egress at the free surfaces, rather than on their multiplication and glide. In this respect, atomistic molecular dynamics (MD) simulations provide a better picture of internal stresses and strains in deformed NPs. Such simulations were performed for faceted single crystalline NPs of several pure metals with face centered cubic (FCC) structure, such as Au⁷, Ni⁸, Pt⁹, and Cu¹⁰. Also, MD simulations of compression of rounded Pd NPs were performed¹¹. The conclusions of these works are similar – the dislocation nucleation is a stochastic process, so that even two neighboring NPs of identical dimensions deformed to the same compressive strain may exhibit different dislocation substructures. In the case of Cu NPs, the dislocation densities in the compressed Cu derived from MD simulations were about $2 \times 10^{15} \text{ m}^{-2}$ in the range of strains of 0.05-0.30, and $1.5 \times 10^{16} \text{ m}^{-2}$ in the range of strains of 0.30-0.65. The densities of geometrically necessary dislocations in compressed Pt NPs determined experimentally employing electron backscattering diffraction mapping were about $1 \times 10^{15} \text{ m}^{-2}$ in the range of strains of 0.05-0.40, and $4 \times 10^{15} \text{ m}^{-2}$ in the range of strains of 0.40-0.80¹². As can be seen, both atomistic MD simulations and experimental studies give the same order of magnitude of dislocation density in deformed FCC NPs in the range of 10^{15} – 10^{16} m^{-2} , depending on plastic strain, and also demonstrate high scatter of these values. Also, two co-authors of the present work (JZ and ER) have recently uncovered that the lattice of the particle systematically rotates upon compressive plastic deformation, and the degree of rotation increases nearly linearly with increasing plastic strain¹³. In the context of the present study this means that the particles deformed to different plastic strains may expose different non-singular surfaces, which may play a certain role in kinetics of hydrogenation. Since no systematic studies of the effects of surface orientation on kinetics of hydrogenation are available in the literature (most studies are limited to low-index singular surfaces¹⁴), we could not employ the knowledge gained in Ref.¹³ in discussion of the results of the present study.

We hope that we convinced the Reviewer that the current state of knowledge only enables us to make some general statements about the dislocation density in deformed particles, and the change of their lattice orientation with increasing plastic strain. To reflect this, we have added the following line to the main text on p. 6:

“Also, we estimate the post-deformation dislocation density in the compressed particles to be of the order of 10^{15} - 10^{16} m⁻² (see SI section 4)”

In addition, a slightly shortened version of the discussion above has been added to the SI under the heading “Estimation of post-compression dislocation density”, which reads as:

“Atomistic molecular dynamics (MD) simulations from the literature can provide an estimate of the dislocation density in the deformed nanoparticles. Such simulations have been performed for faceted single crystalline nanoparticles of several pure metals with face centered cubic (FCC) structure, such as Au³, Ni⁴, Pt⁵, and Cu⁶, and also for compression of rounded Pd particles⁷. The conclusions of these works are similar – the dislocation nucleation is a stochastic process, so that even two neighboring particles of identical dimensions deformed to the same compressive strain may exhibit different dislocation substructures. In the case of Cu particles, the dislocation densities in the compressed particles derived from MD simulations were about 2×10^{15} m⁻² in the range of strains of 0.05-0.30, and 1.5×10^{16} m⁻² in the range of strains of 0.30-0.65. The densities of geometrically necessary dislocations in compressed Pt particles determined experimentally employing electron backscattering diffraction mapping were about 1×10^{15} m⁻² in the range of strains of 0.05-0.40, and 4×10^{15} m⁻² in the range of strains of 0.40-0.80⁸. As can be seen, both atomistic MD simulations and experimental studies give the same order of magnitude of dislocation density in deformed FCC nanoparticles in the range of 10^{15} - 10^{16} m⁻², depending on plastic strain (the plastic strains in this study are on the order of 0.083-0.667, representative of a 5 and 40 nm compression respectively), and also demonstrate high scatter of these values.”

1.4 the state of stress with H2 for which, again, different levels of modelling can be attempted, from simple homogenous imposition of a dilatational eigenstrain (e.g. fictitious thermal expansion coefficient) in a closed form model to more complex FE simulations. The constraint by the substrate is indeed the key in the change of residual stresses. This would give an idea of the magnitude of the stress that can be attained during H2 loading. There are several papers in the literature explaining how to convert H2 fraction (as a function of the phase) into stress, for instance in works on thin Pd films with H2 loading.

Our reply: First, we again emphasize that the stress during hydrogenation in each particle will be individual, as the consequence of several competing factors, i.e., substrate clamping, individual particle morphology (see comment to question 3 further below) and hydrogenation history. Hence, it is not realistic to attempt any description that captures the behavior of each individual particle (and deformation level). However, we can provide a very rough estimate for an averaged response of all particles. We make use of the isotropic Young's elastic modulus of Pd (121 GPa) together with as-of-yet non-peer-reviewed results from our group (10.26434/chemrxiv-2025-btqj2), including in situ AFM measurements of the volume expansion during hydrogenation of Pd nanoparticles very similar to the ones studied here and nanofabricated in the same way.¹⁵ In this AFM study, the hydrogenation resulted in a ~ 11% volume expansion (which is comparable to the 10 % volume expansion reported for bulk Pd¹⁶). Assuming fully isotropic expansion of the hydride nucleus yields an approximate linear hydrogenation stress of 4.4 GPa. We want to emphasize, that beyond a large variation in the particle-to-particle hydrogenation stress levels, which is a consequence of individual particle morphologies and microstructures, we also have additional conditions that could lead to this approximate hydrogenation stress being both under- or overestimation, e.g. plastic relaxation processes during the hydrogenation or increased stress levels due to strong substrate clamping by the sapphire substrate. Consequently, we can only state that the stress levels during hydrogenation are in the GPa range, which is comparable to the stress levels commonly reached during the hydrogenation of annealed and nano-crystalline Pd thin films¹⁷⁻¹⁹, where this level of stress is sufficient to activate dislocation-nucleated plasticity in both the annealed and non-annealed samples. Hence, based on this (admittedly simplified but physically reasonable) analysis, we find that we indeed are in the regime where dislocation-nucleated plasticity is activated, consistent with the main findings of our study.

Furthermore, the main text on p. 18-19 has been updated in the following way:

An estimation of the stress levels induced in the particles upon hydrogenation yields the values of the order of ~1 GPa (see SI section 18). Notably, these values are indeed comparable to the stress levels reached during the hydrogenation of annealed and nano-crystalline Pd thin films, as reported by Delmelle et al.⁴⁶, where this level of stress was sufficient to activate dislocation-mediated plasticity in both annealed and non-annealed samples.

2. Modelling the diffusion process and associated energetics. Here as well, the authors could have a

look at relatively simple (though approximate) models, maybe only to compare extreme cases. One can make assumptions on the diffusion coefficients associated to piping diffusion in dislocations and GB, from literature, and justify the ratio of kinetics with or without dislocations.

Our reply: We agree with the Reviewer that understanding the diffusion process is important and we emphasize that the kinetics enhancing effect of dislocations is generally well-known for hydrogen in metals. To support this general statement, we can first refer to the assessment of our work by Reviewer 2: “The result is a very comprehensive understanding of the processes involved in absorption and desorption of hydrogen in Pd, i.e., from the clamping effect to the formation and annihilation of dislocations and development of strain. Although these phenomena are described earlier this paper shows clear evidence”. For additional literature references, we refer to an *ab initio* calculation study by Kimizuka et al.²⁰ where they show that the diffusion coefficient of hydrogen in Pd increases in the expanded lattice structure normally found around defects. There is also a wealth of experimental studies that demonstrated increased diffusion through dislocation-rich Pd, mainly on thin films, e.g. Delmelle et al.¹⁷ and Kirchheim²¹, as well as the kinetic accelerating effect of grain boundaries in Pd nanoparticles similar to ours.²² Therefore, we argue that it is reasonable to assume that these mechanisms are active in our system, too. As any approximation or calculation would, at best, crudely represent the average behavior of all particles, without accounting for their intrinsic differences in defect density, distribution, and other factors, we consider such efforts unnecessary. Or to put it differently, we do not think we would learn anything really new or more quantitative by attempting such assumptive modelling.

To, nevertheless, strengthen the argument that dislocations indeed enhance H₂ diffusion, we have also added the Kimizuka et al.²⁰ and Kirchheim²¹ references as reference 47-48 on page 13 of the main text. “Having confirmed the initial presence of dislocations, and their hydrogen sorption accelerating effect^{12,44,46-48}”

3. An interesting observation is that specimens with a small dislocation density initially (such as uncompressed NPs) may, under H₂ sorption, lead to a starvation of the dislocations (“starvation” is the word that is used in the dislocation community to express the “cleaning” of a material that get rid off dislocations by mechanical stress). While with a sufficiently high dislocation density, dislocation multiplication mechanisms will start playing a role. I remember papers in Nature Communication or Nature Materials simulating with discrete dislocation dynamics compression of nanopillars and looking at these phenomena. They provide data out of which one can possibly define a threshold dislocation density separating the two regimes, as a function of specimen size.

Our reply: Since our particles exhibit a high variation in yield stress, likely due to a high variation in initial dislocation densities, we estimate the latter by studying the upper and lower limit of the compressive strength of our particles. As an upper limit for the load, F_{crit} , we use the maximum load applied to any single crystalline “strain burst” particle (55 μ N). The lower limit, we estimate from the lowest, clearly visible “step yield” of stair-case yielding particles (~7.5 μ N). To calculate the top area of the particles, A , we use the average diameter (196 nm) from the uncompressed particles in SI Fig 18b. Note that this diameter technically is measured after the hydrogenation cycling process, but it’s close enough to the nominal diameter of the particles (200 nm) that it is still valid. Taken together, the calculated compressive strength $\tau_c = \frac{F_{crit}}{A}$, yields an upper and lower limit of 0.5 and 3.8 GPa respectively, which is in good agreement with the yield strength of similarly sized single crystalline Pd particles fabricated by solid state dewetting.⁶

Next, we apply equation 1 from El-Awady²³

$$\frac{\tau}{G} = \frac{\beta}{d\sqrt{\rho}} + \alpha b\sqrt{\rho} \quad (2)$$

where G is the isotropic shear modulus of Pd (44 GPa), d is the particle diameter, ρ is the dislocation density, b is the Burgers vector ($3.89 \times 10^{-10} \text{ m} \cdot \frac{\sqrt{3}}{2}$ for {111}<011> slip system in Pd) and α and β are dimensionless constants from El-Awady²³. Using eq. (2) and solving numerically for ρ gives dislocation densities either on the order 10^{11} - 10^{12} m^{-2} or 10^{14} - 10^{16} m^{-2} (Figure 1a). Scaling the normalized yield strength $\frac{\tau}{G}$ with \sqrt{d} and the dislocation density ρ with d (Figure 1b), reproduces the generalized size-dependent crystal strength plot of El-Awady²³. Here, we note that our minimum limit of the critical load (7.5 μ N) results in dislocation densities close to the critical density ρ_c , which is the inflection point between the dislocation starvation and forest hardening regime. In fact, if we lower the minimum critical load to 3.4 μ N instead of 7.5 μ N – which is not an unreasonable assumption for a subset of the particles due to the high scatter of yield strengths – then the two dislocation density estimates (Figure 2), one on the dislocation starvation flank and one on the forest hardening flank, overlap at the critical dislocation density ρ_c , i.e. the dislocation density of many of our particles is likely quite close to ρ_c .

Figure 1: Dislocation density estimation. (a) Normalized yield strength as a function of dislocation density ρ using eq. 1. The blue shaded regions are the two regions of ρ that represent the yield strengths of our particles, one region of $\rho \sim 10^{11}$ - 10^{12} m^{-2} in the dislocation starvation regime (left of the critical dislocation density ρ_c), and one region of $\rho \sim 10^{14}$ - 10^{16} m^{-2} in the forest hardening regime (right of the critical dislocation density ρ_c). (b) Scaled normalized yield strength $\tau\sqrt{a}/G$ vs scaled dislocation density ρd^{23} .

Figure 2: Dislocation density calculation using the extreme lower limit. Using $3.4 \text{ } \mu\text{N}$ instead of $7.5 \text{ } \mu\text{N}$ as the limit for the lower yield load results in the two dislocation density estimates (shaded blue regions), one on the dislocation starvation flank and one on the forest hardening flank, to overlap at the critical dislocation density ρ_c .

Taken together, this tells us that we have a significant spread in initial dislocation densities in our particles, and as such a high variation in yield stress, with many particles having dislocation densities close to the critical dislocation density ρ_c . Consequently, we can't from this analysis alone say whether the subgroup of particles with higher yield stress are in the dislocation starvation regime or the forest hardening regime. However, if we compare our results to recent studies on softening of Pd particles⁶, where the hydrogen cycled (soft) particles had a dislocation density around 10^{14} m^{-2} (which consequently is around the value of ρ_c for this size of particles), we argue that it is likely that initial dislocation densities in our particles before hydrogen cycling are lower than for these cycled particles. Also, as stated in our answer to comment 1.3 above, dislocation densities in the range 10^{14} - 10^{16} m^{-2} are expected in the compressed particles. Taken together, the likely initial dislocation densities for our particles are in the 10^{11} - 10^{12} m^{-2} range rather than the 10^{14} - 10^{16} m^{-2} range, which would put our non-deformed particles in the dislocation starvation regime.

A version of the discussion above has been added to the SI under the heading "Estimation of dislocation density post-annealing" and the following line is added on p. 5-6 of the main text

"From the yield strength of the particles, we can estimate that the dislocation density in the non-deformed annealed particles is about 10^{11} - 10^{12} m^{-2} , which would put the annealed (but not deformed) particles in the dislocation starvation regime⁴⁰, where the pre-existing dislocations glide to and annihilate at the surface at increased stress levels, e.g. during hydrogenation (see SI section 3 for more information)."

This is not to say that the authors should pursue all these lines, but the value and impact of the work would increase a lot if some of the experimental findings could be put into equations.

Our reply: We appreciate the Reviewer acknowledging that not all of the requests are necessary or realistic to address within the scope of this work. Accordingly, we have attempted to quantify certain aspects as suggested,

while opting not to do so for others that we deem either irrelevant or beyond the scope of this work and our current capabilities.

Finally, the analysis of the defects needs to go further. The authors are invited to consult the literature on the subject for instance a series of paper by Idrissi and coworkers on defects in Pd films which revealed numerous features including, on top of dislocations, twins, grain growth, production of five folds twins, 9R phase, stacking faults which vary in terms of deposition conditions as well as H₂ sorption levels. In other words, the initial defect structure can be more complex than “only dislocations” and can evolve in a complex manner as well under the very large stress that can be built during H₂ sorption (showing again the need to quantify a little more the magnitude of these stresses).

Our reply: We agree with the Reviewer that the type and density of defects in every individual particle is likely the key to understanding in full detail their individual hydrogen sorption behavior. However, we argue that the strength of this study lies in its statistical analysis of a **multitude** of individual particle groups deformed to different degrees. As the Reviewer suggests, the individual defect structures of every particle will be very complex and would also likely be needed to be captured in its entirety to fully infer the hydrogen sorption characteristics of its parent particle – an undertaking that is unfeasible with today’s state of the art methods for the number of particles studied here. Therefore, we decided to keep the discussion of defect types at a relatively general level and tackle the problem from a statistical point of view instead.

Nonetheless, to take the Reviewer’s comment into account, we have updated the main text in the following ways to reflect that hydrogenation of Pd can lead to many different types of defects (p. 3)...:

“Furthermore, the hydrogenation process itself may alter the dislocation networks, e.g., by creating new dislocations^{13,18-22} or annihilating pre-existing ones^{12,23}. It may also introduce or alter other types of defects, such as stacking faults²⁴ and 9R-phases²⁵. This results in highly complex internal microstructure, where the hydrogenation properties of a hydride forming metal like Pd are influenced not only by the initial strain and defect state, but also by its hydrogenation history”

...but that we will not delve deeper into these (p. 4):

We should emphasize here that, following the discussion above regarding the difficulty of full reconstruction of the individual defect networks of hundreds of nanoparticles, we will keep the discussion of different types of defects non-specific and mainly focus on general deformation trends. We would also like to emphasize that both the stacking faults and the patches of 9R phase are formed as a result of nucleation and propagation of Shockley partial dislocations, so that a general discussion in terms of dislocation substructure is still relevant.

The references

24 Amin-Ahmadi, B. et al. Dislocation/hydrogen interaction mechanisms in hydrided nanocrystalline palladium films. *Acta Materialia* **111**, 253-261 (2016).

and

25 Amin-Ahmadi, B. et al. High resolution transmission electron microscopy characterization of fcc→ 9R transformation in nanocrystalline palladium films due to hydriding. *Applied Physics Letters* **102** (2013).

have also been added to the text.

Reviewer #2 (Remarks to the Author):

The paper contains an extensive analysis of the absorption-desorption behavior of over 300 single Pd nanoparticles (deformed and undeformed) on a sapphire substrate with adequate statistical support of the findings.

The result is a very comprehensive understanding of the processes involved in absorption and desorption of hydrogen in Pd i.e. from the clamping effect to the formation and annihilation of dislocations and development of strain. Although these phenomena are described earlier this paper shows clear evidence. Most interesting are the visualizations of the changes in morphology /shape/protrusions resulting from H-induced deformation mechanisms. The discussion is underpinned by data, the complete data sets in the Supplementary allow verification although the color intensity differences in the figures are sometimes hard to distinguish.

Our reply: We thank the Reviewer for this overall positive assessment of our work.

Comparing the undeformed and deformed particles raises questions at some points. Can the authors argument on these to support their conclusions.

(1) Surface contamination by oxides at fabrication - deformation could damage the oxide layer and enhance H uptake

Our reply: This was indeed one of our initial worries as well. However, we have three arguments why oxides or other surface contaminants should not impact our findings in any significant way. (i) Hydrogenation of metals in general is accompanied by the reduction of surface oxide. In the case of Pd this reduction occurs already at low temperatures^{24,25}, so that after a few cycles the particles should be in a relatively pristine surface state. Indeed, the sorption kinetics speed up over the first hydrogenation cycles (see Figure 2c-e and cycle 1-5 of Figure 3a of the main article), arriving later at the plateau (cycle 6-18 of Figure 3a of the main article), which corroborates the argument of surface oxide reduction. After these initial cycles, all particles – both compressed and non-compressed – exhibit comparable absorption times (see Figure 2e and cycle 5 - 10 in Figure 3a of the main article). The main deacceleration of the non-compressed particles happens after this stage (after cycle 18, see Figure 2f and Figure 3a of the main article). For this deacceleration to be the result of other surface contaminants, it would require (ii) the contaminants would be present on the particles beforehand or (iii) the contaminants are introduced during the measurement. Here we would like to emphasize that all particles studied in our work, both compressed and non-compressed ones, are located on the same substrate and are measured simultaneously. For the above points (ii) and (iii) regarding surface contaminants, this means the following: If contaminants would be present on the sample from the fabrication and be the reason for the drastic deacceleration of the non-compressed particles according to point (ii), this would require these contaminants to suddenly “activate” on the non-compressed particles after around 18 cycles – which we deem highly unlikely. If contaminants on the other hand would be introduced during the measurements according to point (iii), this would mean that said contaminants mainly attach to and affect the non-compressed particles – which we also deem highly unlikely. To further corroborate this point, we have attached a figure below (Figure 3) showing the spatial distribution of the particle absorption times for a late hydrogenation cycle (cycle 69). The compressed particles are highlighted with a dashed white box. This Figure clearly demonstrates that there are no localized, compact regions of “slow” particles (large t_{30-50}) among the non-compressed particles (those outside the white box in Figure 3). Such regions would be expected if non-uniform, localized surface contaminants were present on the non-compressed particles. With all this taken together, we argue that the effect of oxides or other surface contaminants is not significant enough to affect the main findings of this work.

A version of the discussion above has been added to the SI under the heading “Oxide reduction and the effect of other potential contaminants” and the following line is added on p. 9 of the main text

“The accelerated kinetics together with the reduced spread in t_{30-50} is most likely the result of the reduction of surface oxide layers formed upon exposure to ambient conditions after fabrication (for further discussion regarding why oxides or other surface contaminants should not have any significant effect on the results, see SI section 6)”

Figure 3. Spatial distribution of particles with their corresponding t_{30-50} absorption times at hydrogenation cycle 69. The nanocompressed particles are highlighted with a white dashed box. We note that among the non-compressed particles, there are no “slow” areas and the different absorption times are evenly distributed.

(2) on p 12/13 ref 9 and 46 are mentioned to indicate that compression could lower the coherency strain induced hydrogenation barrier - the relation to compressive deformation is unclear. And: The strains in that framework are proportional to the systems volume - and discussed for macroscopic systems - in the case of nanoparticles the surface plays an important role and allows faster relaxation and dislocation annihilation - I doubt if the model is fully applicable to nano-sized systems.

Our reply: We agree with the Reviewer that ref 9 (Schwarz, R. B. & Khachatryan, A. G. Thermodynamics of open two-phase systems with coherent interfaces: Application to metal-hydrogen systems. *Acta Materialia* **54**, 313-323 (2006)) is discussing macroscopic systems. However, in ref 10 (Griessen, R., Strohhfeldt, N. & Giessen, H. Thermodynamics of the hybrid interaction of hydrogen with palladium nanoparticles. *Nature Materials* **15**, 311-317 (2016)), the authors extend the model of Schwarz & Khachatryan to the absorption process in nanosized systems. With regards to ref 46 (Grönbeck, H. & Zhdanov, V. P. Effect of lattice strain on hydrogen diffusion in Pd: A density functional theory study. *Physical Review B* **84** (2011)), the authors specifically discuss their model applied to nanoparticles. As such, we would argue that the main assumption of the model, i.e. that coherency strain is affecting the (absorption) phase transition pressure of Pd, holds at least for single crystalline nanoparticles. Regarding whether the model can be extended to dislocation-containing nanoparticles, we agree with the Reviewer that it may not be fully applicable, which we also acknowledge on pages 14-15 of the main article. However, we still consider this model as a valuable basis for the discussion, and hope that some of the findings here (e.g. that the absorption phase transition pressure is lowered for compressed particles compared to their non-compressed counterparts) can serve as a basis to develop further models of how dislocations and global strain levels affect hydrogen sorption properties of hydride forming nanoparticles.

p.15 Clamping effect: the discussion is about the softness of the substrate but how about the chemical interaction with the substrate - the bonding between the particle and substrate plays a major role. Clamping of the undeformed particles is mentioned to be stronger, why? The surface area of deformed particles in contact with the substrate is expected to be larger - what is the influence of that on the sorption behavior?

Our reply: In our manuscript we discuss the clamping in terms of adhesion energy between Pd particles and substrate. This energy represents a thermodynamic parameter which includes several contributions, chemical bonding being one of them. The clamping stress on the particles from the substrate during hydrogenation is a direct consequence of the strong bonding between the particle and the substrate, resisting the particle's expansion. In regards to the question about why we assume that the clamping is stronger for the non-compressed particles, this is discussed on p. 18-19 of the main article and is based on the observations of shearing of the non-compressed particle close to substrate, as well as the extensive damage to the interface region of the substrate observed by cross-sectional TEM in Figure 4g-i of the main text. Also, as discussed on p. 14 of the main article, external strains such as clamping strains from the substrate are one of the parameters that can affect the hydrogenation phase transition pressure. As such, the higher phase transition pressures exhibited by the non-compressed particles in Figure 3b-c in the main text is consistent with this argument (even though we acknowledge that increased phase-transition pressure of the non-compressed particles can have other contributions as well, as discussed on p. 14-15 of the main article). Furthermore, this argument is corroborated by observations during trial runs of the nanocompression of particles on Si substrate where nanocompression sometimes led to the particles fully detaching from the substrate.

As for the physical reasons for stronger clamping for the non-compressed particle, it may be related to the non-equilibrium character of the particle-substrate interface in the compressed particles. Indeed, it is well-known that intensive plastic deformation of metals leads to dislocation absorption by the grain boundaries and interfaces and significant (up to 50%) increase of their energy.²⁶ The non-equilibrium state of the grain boundaries can even survive recrystallization.²⁷ The increase of Pd-sapphire interface energy in the compressed nanoparticles translates into the corresponding decrease of the adhesion energy.

Regarding the question of whether the compressed particles have increased contact area with the substrate and whether this significantly influences our conclusions, we argue that any such effect would be most pronounced during the initial hydrogenation cycles, that is, immediately after nanoparticle compression. After all hydrogenation cycles, the particles have restructured to the degree that it is essentially impossible to distinguish the compressed particles from the non-compressed ones by SEM and cross-sectional TEM (as discussed in the "Particle morphology analysis" section of the main article). This together with the proposed mechanism of mass transport away from the substrate interface during hydrogenation cycling (and corroborated by the lack of voids in the particle-substrate interface region of a freshly fabricated particle in Figure 1b of the main article), lead us to the argument that any effect related to the particle-substrate contact area should be greatest during the initial hydrogenation cycles. However, the difference in sorption kinetics between compressed and non-compressed particles is the greatest at the later hydrogenation cycles (Figure 3a of the main article), and the difference in absorption plateau pressures is similar between compressed and non-compressed particles over all

hydrogenation cycles (Figure 3b-c of the main article). Taking all this together, we argue that any area-dependent, or any other geometrical / shape effect for that matter, would not be significant enough to alter the main findings of this study.

To better reflect this in the main text, we have updated the text on p. 20 with the following

“As a second aspect, we can also conclude here that the observed differences in t_{30-50} and P_{abs} between compressed and non-compressed particles (cf. Figure 3) are not due to size variations, since these are too small (SI Figure 18). Specifically, after the full hydrogen cycling process, the recrystallisation processes discussed here result in convergence of sizes of the compressed and non-compressed particles, so that the difference in diameters even between the most compressed particles (40 nm) and their non-compressed counterparts is relatively small, i.e., less than 10% (SI Figure 18b). This, combined with the observed hydrogenation behavior (i.e. the fact that the largest difference in kinetics is observed at the latest cycles (Figure 3) without significant overall change in transition pressures) effectively rules out shape or geometric factors as explanations for the observed effects.”

On p22: discussion on the slow down of the H sorption kinetics - the undeformed particles showing the largest slow down; in the paper this is attributed to dislocation annihilation (indeed could be a reason) unclear to me why in the deformed particles the annihilation would be less. Is there any proof for that statement. After many cycles it is expected that the differences in dislocation densities between both type of particles reduces.

Our reply: We agree with the Reviewer that the difference in dislocation densities between the compressed and non-compressed particles is expected to eventually diminish. To highlight this we have added an additional figure to the SI that presents the fit of an exponential model $t_{30-50} = t_0 e^{kx}$, where x is the number of hydrogenation cycle, to the t_{30-50} absorption time data for the non-compressed (0 nm) particles, the 5-10 nm compressed particles (grouped together for additional data points), and the 35-40 nm compressed particles (also grouped together). To make sure any oxides on the surface of the particles are sufficiently reduced (see comment above regarding reduction of oxides), the first cycles included in the fit is cycle 10. For convenience we have presented this figure below (Figure 4). This figure highlights that (i) the particles exhibit an initial absorption time at cycle 10 (t_0 , panel c) that is proportional to their compression degree, and that (ii) the non-compressed particles have a significantly higher de-acceleration rate k compared to the two groups of compressed particles (k , panel d). We attribute this difference to a transition of the compressed particles from a dislocation starvation to a forest hardening-dominated regime (see comment 3 from Reviewer 1 above as well as SI section 3). In other words, the compression introduces so many new dislocations in the particles that it becomes “harder” for them to heal out compared to dislocations in the non-compressed particles, which in turn leads to the different deacceleration rates in compressed and non-compressed particles.

Figure 4. Exponential deacceleration model fit to the experimental data. (a) The model $t_{30-50} = t_0 e^{kx}$ fitted to the non-compressed (0 nm) particles, the 5-10 nm compressed particles (grouped together), and the 35-40 nm compressed particles (also grouped together). (b) Magnified view of the fit of the compressed particle sub-groups in panel a (dashed box). (c-d) The fitted model parameters, t_0 (c) and k (d) for the three different sub-groups.

A version of the above discussion has been to the SI under the heading “Kinetics deacceleration model” and the following line on p. 13 of the main article has been updated to reflect this:

“We hypothesize that such sessile dislocation structures can survive the process of hydrogen cycling (see SI section 13 for more information).“

After clarification and discussion of the above points raised, the study is eligible for publication.

Our reply: We thank the Reviewer for his/her general recommendation and hope to have answered all raised comments satisfactorily.

Reviewer #3 (Remarks to the Author):

In this manuscript, the authors study the kinetics and the structural consequences of hydrogen absorption in Pd nanoparticles that have previously undergone different levels of plastic deformation. The manuscript is of high-quality, very dense, with significant results and is well suited for publication in Nature Communications.

Our reply: We thank the Reviewer for this overall positive assessment of our work.

I agree with most of the conclusions drawn by the authors. However, I would appreciate if the authors could consider the following comments:

- The discussion regarding the influence of dislocation on hydrogen absorption is based on the measurement of the average t_{30-50} absorption times (Figure 3, Supplementary figure 6), the plateau pressures (Supplementary figure 10). In these figures, the error bars represent the standard error. The variations of t_{30-50} and the plateau pressures vs cycle cannot be considered as significant as all error bars overlap (except for the evolution of t_{30-50} for the uncompressed particles compared with that for the compressed ones). I think the authors should include a statistical analysis to support their conclusions.

Our reply: We thank the Reviewer for this comment but want to clarify that the error bars in Figure 3 (and in fact in the entire manuscript) represent the standard deviation of the presented parameter over the corresponding individual particle population. Hence it is **not** a representation of an error but of the spread within a population of individual particles, as clearly stated in the captions of Figures 2 and 3 of the main text. Still, we agree with the Reviewer that additional statistical analysis is beneficial to support our conclusions. Therefore, we have performed a Student's t -test on the t_{30-50} kinetics absorption times on the two extremes of the compressed particles, i.e. the 5-10 nm compressed particles (grouped together for improved statistics), and the 35-40 nm compressed particles (also grouped together for improved statistics). The null hypothesis for this statistical t -test is that the data points are independent random samples from normal distributions with equal means and equal but unknown variances. The alternative hypothesis is that the data comes from populations with unequal means. We have added an additional figure to the SI which shows for which cycles the test rejects the null hypothesis at the 5% significance level (highlighted with filled squares at the corresponding data points). For convenience we present this figure below (Figure 5). This test shows that the difference in absorption kinetics indeed is significant between the two extremes of the compressed particles (i.e. the 5-10 nm and the 35-40 nm compressed particles), at least for the first ~50 cycles.

Figure 5. Student's *t*-test hypothesis testing on the two extremes of the compressed particles, i.e. the 5-10 nm compressed particles, and the 35-40 nm compressed particles. For cycles where the null hypothesis, i.e. that the kinetics of the 5-10 nm and the 35-40 nm compressed particles are represented with the same distribution with equal means, is rejected at the 95% confidence level is highlighted with filled squares

We have also performed the same Student's *t*-test on the absorption plateau pressure data. Here, we compared all three compression groups of Figure 3b-c of the main text with each other (i.e. comparing the 0 nm compressed particles to the 5-20 nm compressed particles, comparing the 0 nm compressed particles to the 25-40 nm compressed particle and finally also comparing the 5-20 nm compressed particles to the 25-40 nm compressed particle). We have also added this comparison as an additional figure to the SI which shows for which cycles the test rejects the null hypothesis at the 5% significance level (highlighted with filled squares at the corresponding data points). For convenience we also present this figure below (Figure 6).

Figure 6. Student's *t*-test hypothesis testing on the absorption plateau pressures for three groups of differently compressed particles; (a) the 0 nm compressed particles compared to the 5-20 nm compressed particles, (b) the 0 nm compressed particles compared to the 25-40 nm compressed particles and (c) the 5-20 nm compressed particles compared to the 25-40 nm compressed particles. For cycles where the null hypothesis, i.e. that the absorption plateau pressures are represented with the same distribution with equal means, is rejected at the 95% confidence level is highlighted with filled squares.

From this analysis it is clear that there is a statistically significant difference between the absorption plateau pressures for the non-compressed particles and both groups of compressed particles. However, whether there is a statistically significant difference between the absorption plateau pressures between the two groups of

compressed particles is less clear. Therefore, we have updated the main text on page 14-15 to reflect this as follows:

“Additionally, comparing the least compressed group (5-20 nm compression) to the most compressed group (25-40 nm compression) apparently reveals that the greater the compression level, the greater reduction in P_{abs} . However, this difference is not statistically significant (see **SI section 14** for a statistical analysis).”

In addition, a version of the discussion above has been added to the SI under the heading “Statistical analysis of hydrogenation kinetics and isothermal plateau pressures”.

- Line 269: “existing dislocations can reach the surface and shear the crystal upon hydrogenation”. Is the behavior of monocrystalline nanoparticles different from that of polycrystalline ones, which contain not only dislocations but also grain boundaries?

Our reply: With regards to the shearing of the particles, from the SEM figures of Figure S1 in SI, we provide a larger version below (Figure 7) of the four particles that have sheared (highlighted with white arrows) due to hydrogenation. For the particles that initially included grain boundaries (panel a-d) the shearing follows the grain boundaries, as would be expected. Indeed, the lattice dislocation can be absorbed by the grain boundary and dissociate there in grain boundary dislocations; the glide of the latter along the grain boundary causes the particle shearing along the boundary plane. For the single crystalline particles, the particles appear to shear at more arbitrary crystal planes, or more specifically, from the top-side view SEM micrographs discussed here, it is difficult to ascertain the exact shearing planes. To this end, PXRD analysis reveals that the particles have a preferred crystal orientation of $[111]$ normal to the substrate surface (see Figure 8 pole figures for the (111) , (200) and (220) peaks which align well with the modelled 111 preferred orientation). This means that the shearing plane in Figure 4h of the main text is a $\{111\}$ plane, as expected for FCC crystals such as Pd. The shearing visible in the particles of Figure S1 in SI is difficult to identify from the top-side view SEM image. It can be associated with one of the three $\{111\}$ planes inclined with respect to the substrate, or with the $\{110\}$ planes. The latter represent an anomalous slip plane for FCC crystals, but has been observed in nano and microcrystals where they have been attributed to nucleation of defects at the surface of the nanostructures.²⁸

To summarize and try to answer the Reviewer’s question in a more concise form, for single crystalline particles, we find it likely that if shearing occurs, it would be on either a $\{111\}$ slip plane or the anomalous $\{110\}$ slip plane, while for polycrystalline particles, the shearing will likely also occur along a grain boundary (which not necessarily excludes activation of any of the two already mentioned slip systems).

Figure 7. SEM micrographs of individual Pd nanoparticles before (a,c,e,g) and after (b,d,f,h) hydrogen cycling for polycrystalline particles with apparent grain boundaries (a-d) and single crystalline particles (e-h) that have sheared during the hydrogenation process. The imaged sample (S2) was fabricated in the same way as the sample discussed in the main article (S1) - except that no particles have been compressed. The particles were subjected to 19 hydrogenation cycles using the same hydrogenation procedure as for the sample discussed in the main article (S1), see supplementary file Hydrogenation summary - sample 1 and Hydrogenation summary - sample 2. White arrows denote the shearing direction of the individual grains of the particles. Scale bars are 100 nm.

Figure 8. Pole figures for the (111) (a-b), (200) (c-d) and (220) (e-f) Pd peaks – measured (a,c,e) and modelled (b,d,f) based on the hypothesized [111] orientation of the Pd nanoparticles.

- Figure 4d-g: the authors conclude from the bright-field TEM images that “significant dislocation structures appear(...) as regions of dark contrast in both BF and HAADF micrographs”. It is also said that “Dark contrast features in the HAADF micrographs can also be interpreted as local voids or low - density regions inside the particles”.

- The camera length used for HAADF imaging is not given, so it is difficult to evaluate whether dislocations can be observed using this imaging mode.

- If we consider that the images reveal mass-density contrasts as expected in HAADF, the features should rather be voids or low-density regions. As the contrasts in the BF images are similar to those observed in HAADF, it is difficult to draw a clear conclusion about dislocation

Our reply: Camera length was 94 mm at 300kV (which has been added to Methods). After careful inspection of the BF and HAADF TEM micrographs, we agree with the Reviewer that they are too similar to warrant such a strong claim of “significant dislocation structures” apparent in the images. Therefore, we have updated the text to reduce the level of certainty regarding this point.

“The bright field (BF) images furthermore confirm the presence of defects inside the crystals, revealed as networks of darker contrast features in the BF images (Figure 4d, g). The (dark contrast) defect features in the High-Angle Annular Dark-Field (HAADF) images confirms that many of these defects are likely associated with local voids or low-density regions inside the particles (Figure 4e,h)”

As minor comments:

- Supplementary figure 1: the SEM images of the nanoparticles are very small. It is very difficult to analyze in detail whether the particles are considered as single crystalline or polycrystalline.

Moreover, the particles are numbered in Supplementary figure 2, but the numbers are not given in Supplementary figure 1. Please note that a few scale bars are also missing.

Our reply: Supplementary Figure 1 has been updated with larger versions of the SEM images. For Supplementary Figure 2, we want to clarify that the particles in this figure are from the sample of the main article, while the particles imaged in Supplementary Figure 1 are from a secondary sample. The reason for using a secondary sample for the SEM images of Supplementary Figure 1 was to limit any surface contamination, such as SEM carbon deposition, before the first hydrogenation exposures of the main sample.

- Figure 1b: a few dislocations are indicated by white and red arrows. It is difficult to distinguish them from all other contrast variations within the particle. In order to better see the dislocations, could the authors provide Weak Beam Dark Field images (if the used this imaging mode)? Moreover, could they provide an indexation of the highlighted dislocations?

Our reply: Unfortunately, we are not able to provide Weak Beam Dark Field images nor indexing of the dislocations. What we can provide is higher magnified versions of the TEM images found in the main article (Figure 9, Figure 10, Figure 11a) as well as a HAADF image of the as-fabricated particle of Figure 1b of the main article (Figure 11b).

We also refer to the answer to the comment from Reviewer 1 regarding further analysis of the defects, where we argue the strength of this study mainly lies in the statistical analysis of a **multitude** of individual particle groups deformed to different degrees. As the Reviewer suggests, the individual defect structures of every particle will be very complex and would also likely be needed to be captured in its entirety to fully infer the hydrogen sorption characteristics of its parent particle – an undertaking that is unfeasible with today's state of the art methods for the number of particles studied here. Therefore, we decided to keep the discussion of defect types at a relatively general level and tackle the problem from a statistical point of view instead.

We hope that, even though we were unable to characterize the individual defects in the TEM cross sections, we have satisfactorily addressed the comments raised.

Figure 9. (a) Bright-field (BF) and (b) High-Angle Annular Dark-Field (HAADF) TEM micrographs of a nominally 30 nm compressed nanoparticle. Scale bars are 50 nm.

Figure 10. (a) Bright-field (BF) and (b) High-Angle Annular Dark-Field (HAADF) TEM micrographs of a non-compressed nanoparticle. Scale bars are 50 nm.

Figure 11. (a) Bright-field (BF) and (b) High-Angle Annular Dark-Field (HAADF) TEM micrographs of an as-fabricated particle prior to the first H_2 exposure. Scale bars are 20 nm.

- Figure 1j-k: in the literature, monocrystalline nanoparticles have already shown stepwise-staircase curves. Why do the authors say that such curve is characteristic of polycrystalline particles?

Our reply: We agree with the Reviewer that under certain circumstances single crystalline particles have been shown to exhibit staircase load-displacement curves. However, according to the current state of knowledge, the vast majority of single crystalline nanoparticles of pure metals obtained by solid state dewetting exhibit the load-displacement curves with a strain burst. In this respect, we would like to emphasize that the Pd particles studied in our work were annealed at 500 °C for 2 h after fabrication. This temperature is sufficiently high to annihilate most of the defects in the as-fabricated particles. Therefore, we argue that the group of particles that don't exhibit stair-case curves, but rather strain-burst characteristics, are devoid of dislocation sources, and that the majority of such particles are single crystalline. Nevertheless, by stating that we have both particles that exhibit strain burst and staircase yielding, we want to highlight that our initial particles are heterogeneous in terms of morphology/defect density and dislocation sources. The caption for Figure 1 has been updated with the following to reflect this:

"The particle in (c) is a defect-lean single crystal whereas the particles in (d) and (e) are either polycrystals or defect-rich single crystals."

- Figure 2f: why is the error bar far larger for uncompressed particles than compressed ones?

Our reply: The error bars represent one standard deviation of the absorption times, calculated from the individual absorption times of each particle within a given compression-level subgroup. Accordingly, Figure 2f of the main article, which presents the absorption times for all compression groups during hydrogenation cycle 18, highlights the point at which the distribution of absorption times for the non-compressed particles begins to noticeably broaden and deviate from that of the compressed particles. This is further discussed in the "Single particle

statistics” section and Figure 5.

- Line 279: “Importantly, such sessile dislocation structures can survive the process of hydrogen cycling”. Why? Does a reference support this assertion?

Our reply: As microcompression can yield immobile dislocation structures that can survive further compressive stress, we argue that the same structures should also be able to survive (up to) comparable stresses from hydrogenation. However, we agree with the Reviewer that the sentence in its current form is too strong and have reduced the certainty level accordingly:

“We hypothesize that such sessile dislocation structures can survive the process of hydrogen cycling.”

- Line 401: “This void formation is likely caused by the migration of Pd atoms (...) towards the surface of the particles, where they form the observed protrusions”. Why are the protrusions not close to grain boundaries in polycrystalline particles?

Our reply: From the SEM figures of Figure S1 in SI (which will also be provided at higher magnification, see earlier comment), we would argue that most protrusions on polycrystalline particles appear in connection with grain boundaries. Many times, they follow the grain boundaries (panel a-c in Figure below, protrusion edges are highlighted with white arrows). Despite this, we acknowledge that it is unexpected that the protrusions in many instances also do not follow the grain boundaries but instead seem to have grown at an angle to them (panel d-f in Figure below, protrusion edges are highlighted with white arrows). However, we can't follow the growth of the protrusions in-situ with our methods, and we also can't see the initial dislocation structure beneath the surface. Hence, it is well possible that the Pd atoms have migrated along the cores of dislocations that terminate at the grain boundary, resulting in the diffusion flow to the surface that yields the protrusion.²⁹ This would result in a protrusion that appears to “grow” at an angle to the grain boundary. Moreover, the internal stresses in the particles caused by hydrogenation can induce grain boundary migration. Thus, the protrusion that forms at the original grain boundary position can get disconnected from the boundary once the latter migrates to a new position after several hydrogenation cycles. A somewhat similar phenomenon of “ghost lines” – the traces marking the intermittent positions of migrating grain boundary on the sample surface which are disconnected from the final boundary location – is well-known in Physical Metallurgy.^{30,31} To reflect this, we have added the following line on p. 16 of the main text:

“Here we should also add that protrusions that appear to grow at an angle to grain boundaries, i.e. not along the boundary, could stem from internal dislocations that terminated at the grain boundary, and that these dislocations then acted as fast diffusion paths to the surface that yield the protrusion. Moreover, the grain boundaries in the particles may have migrated under the action of hydrogenation-induced internal stresses. The protrusion that forms at the original boundary position is then getting disconnected from the final position of migrated grain boundary.”

Figure 12. SEM micrographs of individual Pd after hydrogen cycling for polycrystalline particles with protrusions growing along (a-c) or at an angle to (d-f) grain boundaries. The imaged sample (S2) was fabricated in the same way as the sample discussed in the main article (S1) - except that no particles have been compressed. The particles were subjected to 19 hydrogenation cycles using the same hydrogenation procedure as for the sample discussed

in the main article (S1), see supplementary file Hydrogenation summary - sample 1 and Hydrogenation summary - sample 2. White arrows denote the shearing direction of the individual grains of the particles. Scale bars are 100 nm.

- Line 242: I think “size” would be more appropriate than “shape”.

Our reply: We are not quite sure to what sentence this comment refers to, as the word “shape” is used for the first time on line 83, and then not again until line 424. We assume the latter is the one that the Reviewer had in mind, and have changed it to “size”

- Line 428-430: “Upon compression, the particle shape becomes more disk-like... on hydrogenation kinetics”. I do not see the link between this sentence and the previous ones.

Our reply: We agree with the Reviewer that this sentence is a bit out of place with the rest of the section. The main point of this section of the main article is to argue that any geometrical / shape effects on the hydrogenation properties (kinetics and phase transition plateau pressure), if significant, should have the greatest impact at the beginning of the hydrogenation process, i.e. right after the particles have been compressed. However, our experiments reveal the opposite trend in kinetics: the difference between compressed and non-compressed particles is most pronounced at the end of the hydrogenation process (see Figure 3a of the main article). At these late stages, the compressed particles have recrystallized to the degree where there is almost no noticeable shape difference between the compressed and non-compressed particles. This effectively should eliminate any shape or geometrical effect as an explanation for the observed results. The same argument also holds for the phase transition plateau pressure, but with the condition that the observed pressures instead have **not** significantly changed during the hydrogenation process. The pressure instead fluctuates sinusoidally as a function of time spent in the hydride phase (see Figure 3b of the main text). As such, we have updated this section (p 20) of the main text in the following way to clarify this (see also answer to question about substrate clamping effects by Reviewer 2):

“As a second aspect, we can also conclude here that the observed differences in t_{30-50} and P_{abs} between compressed and non-compressed particles (cf. **Figure 3**) are not due to size variations, since these are too small (**SI Figure 18**). Specifically, after the full hydrogen cycling process, the recrystallisation processes discussed here result in convergence of sizes of the compressed and non-compressed particles, so that the difference in diameters even between the most compressed particles (40 nm) and their non-compressed counterparts is relatively small, i.e., less than 10% (**SI Figure 18b**). This, combined with the observed hydrogenation behavior (i.e. the fact that the largest difference in kinetics is observed at the latest cycles (**Figure 3**) without significant overall change in transition pressures) effectively rules out shape or geometric factors as explanations for the observed effects.”

- Lines 450-452: “As the nucleation of dislocations in the non-compressed particles is highly stochastic...”. Please explain, this sentence is unclear.

Our reply: The microcompression tests performed on defect-lean nanoparticles and micropillars have firmly established that the nucleation of dislocations is highly stochastic, which means a high spread in nucleation stress, the random nature of nucleation sites, and the large variations in the resulting dislocation substructures. The level of internal stresses in the non-compressed particles developing during hydrogenation cycling is comparable to their yield strength, and, therefore, the resulting dislocation substructure should vary significantly from particle to particle. We then argue that this can explain why the non-compressed particles exhibit such large spread in absorption times, see Figure 5d of the main article. We have updated the text in the following way to clarify this:

“It is well established that the nucleation of dislocations in pristine or dislocation-starved nanoparticles at the onset of plasticity is highly stochastic^{37,67}. The level of internal stresses developing in the non-compressed particles during hydrogenation cycling process is comparable to their ultimate strength.²¹ Therefore, we conjecture that the dislocation substructure in these particles develops in a highly stochastic manner, depending on their individual initial dislocation state and their (stochastic) dislocation nucleation history. The resulting variability of dislocation substructures between the individual particles may indeed explain the increase of t_{30-50} spread upon cycling seen in **Figure 5d**.”

The references

37 Lee, S.-W., Mordehai, D., Rabkin, E. & Nix, W. D. Effects of focused-ion-beam irradiation and prestraining on the mechanical properties of FCC Au microparticles on a sapphire substrate. *Journal of Materials Research* **26**, 1653-1661 (2011)

and

67 Ryu, I., Cai, W., Nix, W. D. & Gao, H. Stochastic behaviors in plastic deformation of face-centered cubic micropillars governed by surface nucleation and truncated source operation. *Acta Materialia* **95**, 176-183 (2015)

have also been added to the text.

- Several figures from the Supplementary file are not discussed in the manuscript. For instance, I did not understand what they bring (for instance the graphs showing PH2 in function of the average dark-field scattering intensity). Also in Supplementary figures 20 and 21, linear fits are given. In several cases, a statistical analysis (linear regression) may indicate that the slopes are not significantly different from 0. Please check.

Our reply: We agree with the Reviewer that some figures in the Supplementary file do not contribute directly to the current discussion. However, our study demonstrates that the microstructure evolution of defect-containing Pd nanoparticles during hydrogenation is quite complicated, so that the models we propose in our discussion of the results will likely need to be expanded or revised in future studies. As such, the included figures may be relevant for other researchers and future studies, which is why we would prefer to keep them in the Supplementary Information.

Regarding the linear fits in Supplementary Figure 20 (now 26) and 21 (now 27), the intention behind the linear fits was to have them more as a guide to the eye and not necessarily imply a claim of a linear dependence. However, we acknowledge that this was not clear and have therefore removed the linear fits in Supplementary Figures 20 and 21 (now 26 and 27).

- 1 Gao, H., Zhang, L., Nix, W., Thompson, C. & Arzt, E. Crack-like grain-boundary diffusion wedges in thin metal films. *Acta materialia* **47**, 2865-2878 (1999).
[https://doi.org/10.1016/S1359-6454\(99\)00178-0](https://doi.org/10.1016/S1359-6454(99)00178-0)
- 2 Levi, M., Bisht, A. & Rabkin, E. Diffusion-induced recrystallization during the early stages of solid-state dewetting of Ni-Pt bilayers. *Acta Materialia* **225**, 117537 (2022).
<https://doi.org/10.1016/j.actamat.2021.117537>
- 3 Dutta, B. & Dayal, B. Lattice constants and thermal expansion of palladium and tungsten up to 878 C by X-ray method. *physica status solidi (b)* **3**, 2253-2259 (1963).
<https://doi.org/10.1002/pssb.19630031207>
- 4 Kozawa, T. *et al.* Thermal stress in GaN epitaxial layers grown on sapphire substrates. *Journal of applied physics* **77**, 4389-4392 (1995). <https://doi.org/10.1063/1.359465>
- 5 Gale, W. & Totemeier, T. Elastic properties, damping capacity and shape memory alloys. *Smithells Metals Reference Book, Eighth Edition; Gale, WF, Totemeier, TC, Eds*, 1-45 (1992).
- 6 Zimmerman, J. *et al.* Drastic softening of Pd nanoparticles induced by hydrogen cycling. *Scripta Materialia* **253** (2024). <https://doi.org/10.1016/j.scriptamat.2024.116304>
- 7 Mordehai, D. *et al.* Size effect in compression of single-crystal gold microparticles. *Acta Materialia* **59**, 5202-5215 (2011). <https://doi.org/10.1016/j.actamat.2011.04.057>
- 8 Sharma, A., Hickman, J., Gazit, N., Rabkin, E. & Mishin, Y. Nickel nanoparticles set a new record of strength. *Nature Communications* **9**, 4102 (2018). <https://doi.org/10.1038/s41467-018-06575-6>
- 9 Padilla Espinosa, I. M. *et al.* Platinum nanoparticle compression: Combining in situ TEM and atomistic modeling. *Applied Physics Letters* **120** (2022). <https://doi.org/10.1063/5.0078035>
- 10 Liang, Z. *et al.* Ultimate compressive strength and severe plastic deformation of equilibrated single-crystalline copper nanoparticles. *Acta Materialia*, 120101 (2024).
<https://doi.org/10.1016/j.actamat.2024.120101>
- 11 Bian, J., Yang, L., Yuan, W. & Wang, G. Influence of hydrogenation on the mechanical properties of Pd nanoparticles. *RSC advances* **11**, 3115-3124 (2021).
<https://doi.org/10.1039/D0RA08974E>

- 12 Zimmerman, J. & Rabkin, E. Nanoparticle recrystallization: kinetics and size-dependent behavior. *Acta Materialia*, 121028 (2025). <https://doi.org/10.1016/j.actamat.2025.121028>
- 13 Zimmerman, J. & Rabkin, E. Sculpturing metal nanoparticles by controlled massive deformation. *Scripta Materialia* **252**, 116248 (2024). <https://doi.org/10.1016/j.scriptamat.2024.116248>
- 14 Sytwu, K. *et al.* Visualizing facet-dependent hydrogenation dynamics in individual palladium nanoparticles. *Nano Letters* **18**, 5357-5363 (2018). <https://doi.org/10.1021/acs.nanolett.8b00736>
- 15 Carmiel-Kostan, M. *et al.* Nanoscale Analysis of Sulfur Poisoning Effects on Hydrogen Sorption in Single Pd Nanoparticles. *ChemRxiv* (2025). <https://doi.org/10.26434/chemrxiv-2025-btqj2>
- 16 Manchester, F., San-Martin, A. & Pitre, J. The H-Pd (hydrogen-palladium) system. *Journal of phase equilibria* **15**, 62-83 (1994). <https://doi.org/10.1007/BF02667685>
- 17 Delmelle, R. *et al.* Effect of structural defects on the hydriding kinetics of nanocrystalline Pd thin films. *International Journal of Hydrogen Energy* **40**, 7335-7347 (2015). <https://doi.org/10.1016/j.ijhydene.2015.04.017>
- 18 Wagner, S. *et al.* Mechanical stress and stress release channels in 10–350 nm palladium hydrogen thin films with different micro-structures. *Acta Materialia* **114**, 116-125 (2016).
- 19 Greenbaum, Y., Barlam, D., Mintz, M. & Shneck, R. Elastic fields generated by a semi-spherical hydride particle on a free surface of a metal and their effect on its growth. *Journal of alloys and compounds* **509**, 4025-4034 (2011).
- 20 Kimizuka, H., Ogata, S. & Shiga, M. Mechanism of fast lattice diffusion of hydrogen in palladium: Interplay of quantum fluctuations and lattice strain. *Physical Review B* **97**, 014102 (2018). <https://doi.org/10.1103/PhysRevB.97.014102>
- 21 Kirchheim, R. Hydrogen solubility and diffusivity in defective and amorphous metals. *Progress in Materials Science* **32**, 261-325 (1988). [https://doi.org/10.1016/0079-6425\(88\)90010-2](https://doi.org/10.1016/0079-6425(88)90010-2)
- 22 Alekseeva, S. *et al.* Grain-growth mediated hydrogen sorption kinetics and compensation effect in single Pd nanoparticles. *Nature Communications* **12**, 5427 (2021). <https://doi.org/10.1038/s41467-021-25660-x>
- 23 El-Awady, J. A. Unravelling the physics of size-dependent dislocation-mediated plasticity. *Nature communications* **6**, 5926 (2015). <https://doi.org/10.1038/ncomms6926>
- 24 Belousov, V., Vasylyev, M., Lyashenko, L., Vilkova, N. Y. & Nieuwenhuys, B. The low-temperature reduction of Pd-doped transition metal oxide surfaces with hydrogen. *Chemical Engineering Journal* **91**, 143-150 (2003). [https://doi.org/10.1016/S1385-8947\(02\)00147-X](https://doi.org/10.1016/S1385-8947(02)00147-X)
- 25 Musket, R. Effects of contamination on the interaction of hydrogen gas with palladium: a review. *Journal of the Less Common Metals* **45**, 173-183 (1976). [https://doi.org/10.1016/0022-5088\(76\)90265-4](https://doi.org/10.1016/0022-5088(76)90265-4)
- 26 Valiev, R. Z., Islamgaliev, R. K. & Alexandrov, I. V. Bulk nanostructured materials from severe plastic deformation. *Progress in materials science* **45**, 103-189 (2000).
- 27 Zimmerman, J., Sharma, A., Divinski, S. & Rabkin, E. Relative grain boundary energies in ultrafine grain Ni obtained by high pressure torsion. *Scripta Materialia* **182**, 90-93 (2020).
- 28 Richard, M.-I. *et al.* Anomalous glide plane in platinum nano- and microcrystals. *ACS nano* **17**, 6113-6120 (2023). <https://doi.org/10.1021/acs.nano.3c01306>
- 29 Legros, M., Dehm, G., Arzt, E. & Balk, T. J. Observation of giant diffusivity along dislocation cores. *Science* **319**, 1646-1649 (2008).
- 30 Mullins, W. W. The effect of thermal grooving on grain boundary motion. *Acta metallurgica* **6**, 414-427 (1958).
- 31 Rabkin, E., Amouyal, Y. & Klinger, L. Scanning probe microscopy study of grain boundary migration in NiAl. *Acta materialia* **52**, 4953-4959 (2004).